# Risk Bounds for Over-parameterized Maximum Margin Classification on Sub-Gaussian Mixtures

**Yuan Cao**
Department of Statistics & Actuarial Science
Department of Mathematics
The University of Hong Kong
yuancao@hku.hk

**Quanquan Gu**
Department of Computer Science
University of California, Los Angeles
Los Angeles, CA 90095, USA
qgu@cs.ucla.edu

**Mikhail Belkin**
Halıcıoğlu Data Science Institute
University of California San Diego
La Jolla, CA 92093, USA
mbelkin@ucsd.edu

## Abstract

Modern machine learning systems such as deep neural networks are often highly over-parameterized so that they can fit the noisy training data exactly, yet they can still achieve small test errors in practice. In this paper, we study this "benign overfitting" phenomenon of the maximum margin classifier for linear classification problems. Specifically, we consider data generated from sub-Gaussian mixtures, and provide a tight risk bound for the maximum margin linear classifier in the over-parameterized setting. Our results precisely characterize the condition under which benign overfitting can occur in linear classification problems, and improve on previous work. They also have direct implications for over-parameterized logistic regression.

## 1 Introduction

In modern machine learning, complex models such as deep neural networks have become increasingly popular. These complicated models are capable of fitting noisy training data sets, while at the same time achieving small test errors. In fact, this *benign overfitting* phenomenon is not a unique feature of deep learning. Even for kernel methods and linear models, [5] demonstrated that interpolators on the noisy training data can still perform near optimally on the test data. A series of recent works [4, 20, 11, 2] theoretically studied how over-parameterization can achieve small population risk.

In particular in [2] the authors considered the setting where the data are generated from a ground-truth linear model with noise, and established a tight population risk bound for the minimum norm linear interpolator with a matching lower bound. More recently, [23] further studied benign overfitting in ridge regression, and established non-asymptotic generalization bounds for over-parametrized ridge regression. They showed that those bounds are tight for a range of regularization parameter values. Notably, these results cover arbitrary covariance structure of the data, and give a nice characterization of how the spectrum of the data covariance matrix affects the population risk in the over-parameterized regime.

Very recently, benign overfitting has also been studied in the setting of linear classification [6, 19, 25]. Specifically, [19] studied the setting where the data inputs are Gaussian and the labels are generated

35th Conference on Neural Information Processing Systems (NeurIPS 2021).

from a ground truth linear model with label flipping noise, and showed equivalence between the hard-margin support vector machine (SVM) solution and the minimum norm interpolator to study benign overfitting. [6, 25] studied the benign overfitting phenomenon in sub-Gaussian/Gaussian mixture models and established population risk bounds for the maximum margin classifier. [6] leveraged the *implicit bias* of gradient descent for logistic regression [22] to establish the risk bound. [25] established an equivalence result between classification and regression for isotropic Gaussian mixture models. While these results have offered valuable insights into the benign overfitting phenomenon for (sub-)Gaussian mixture classification, they still have certain limitations. Unlike the results in the regression setting where the eigenvalues of the data covariance matrix play a key role, the current results for Gaussian/sub-Gaussian mixture models do not show the impact of the spectrum of the data covariance matrix on the risk.

In this paper, we study the benign overfitting phenomenon in a general sub-Gaussian mixture model that covers both the isotropic and anisotropic settings, where the $d$-dimensional features from two classes have the same covariance matrix $\boldsymbol{\Sigma}$ but have different means $\boldsymbol{\mu}$ and $-\boldsymbol{\mu}$ respectively. We consider the over-parameterized setting where $d$ is larger than the sample size $n$, and prove a risk bound for the maximum margin classifier. We show that under certain conditions on eigenvalues of $\boldsymbol{\Sigma}$, the mean vector $\boldsymbol{\mu}$ and the sample size $n$, the maximum margin classifier for this problem is identical to the minimum norm interpolator. We then utilize this result to establish a tight population risk bound of the maximum margin classifier. Our result reveals how the eigenvalues of the covariance matrix $\boldsymbol{\Sigma}$ affect the benign property of the classification problem, and is tighter and more general than existing results on sub-Gaussian/Gaussian mixture models. The contributions of this paper are as follows:

- We establish a tight population risk bound for the maximum margin classifier. Our bound works for both the isotropic and anisotropic settings, which is more general than existing results in [6, 25]. When reducing our bound to the setting studied in [6], our result gives a bound $\exp(-\Omega(n\|\boldsymbol{\mu}\|_2^4/d))$, where $n$ is the training sample size. Our bound is tighter than the risk bound $\exp(-\Omega(\|\boldsymbol{\mu}\|_2^4/d))$ in [6] by a factor of $n$ in the exponent. Our result also gives a tighter risk bound than that in [25][1] in the so-called "low SNR setting": our result suggests that $\|\boldsymbol{\mu}\|_2^4 = \omega(d/n)$ suffices to ensure an $o(1)$ population risk, while [25] requires $\|\boldsymbol{\mu}\|_2^4 = \omega((d/n)^{3/2})$.

- We establish population risk lower bounds achieved by the maximum margin classifier under two different settings. In both settings, the lower bounds match our population risk upper bound up to some absolute constants. This suggests that our population risk bound is tight.

- Our analysis reveals that for a class of high-dimensional anisotropic sub-Gaussian mixture models, the maximum margin linear classifier on the training data can achieve small population risk under mild assumptions on the sample size $n$ and mean vector $\boldsymbol{\mu}$. Specifically, suppose that the eigenvalues of $\boldsymbol{\Sigma}$ are $\{\lambda_k = k^{-\alpha}\}_{k=1}^d$ for some parameter $\alpha \in [0, 1)$, and treat the sample size $n$ as a constant. Then our result shows that to achieve $o(1)$ population risk, the following conditions on $\|\boldsymbol{\mu}\|_2$ suffice:

$$\|\boldsymbol{\mu}\|_2 = \begin{cases} \omega(d^{1/4-\alpha/2}), & \text{if } \alpha \in [0, 1/2), \\ \omega((\log(d))^{1/4}), & \text{if } \alpha = 1/2, \\ \omega(1). & \text{if } \alpha \in (1/2, 1). \end{cases}$$

  More specifically, when $\alpha = 1/2$, the condition on the mean vector $\boldsymbol{\mu}$ only has a logarithmic dependency on the dimension $d$, and when $\alpha \in (1/2, 1)$, the condition on $\boldsymbol{\mu}$ for benign overfitting is dimension free.

- Our proof of the population risk bound introduces some tight intermediate results, which may be of independent interest. Specifically, our proof utilizes the polarization identity to establish equivalence between the maximum margin classifier and the minimum norm interpolator. This is, to the best of our knowledge, the first equivalence result between classification and regression for anisotropic sub-Gaussian mixture models.

**Additional Related Work** Our study is closely related to the phenomenon of double descent studied in recent works. [3, 4] showed experimental results and provided theoretical analyses on some

---

[1]After we posted our paper on arXiv, we noticed that the authors of [25] updated their paper a week later to include the results for mixture of anisotropic Gaussians and sharpened their bounds. Our results are still more general as we covered the mixture of anisotropic sub-Gaussian, and we also proved a matching lower bound.

specific models to demonstrate that the risk curve versus over-parameterization has a double descent shape. These results can therefore indicate that over-parameterization can be beneficial to achieve small test risk. [11, 26] studied the double descent phenomenon in linear regression under the setting where the dimension $d$ and sample size $n$ can grow simultaneously but have a fixed ratio, and showed that the population risk exhibits a double descent curve with respect to the ratio. More recently, [17, 15, 18] further extended the setting to random feature models and studied double descent when the sample size, data dimension and the number of random features have fixed ratios.

Our work is also related to the studies of implicit bias, which analyze the impact of training algorithms when the over-parameterized models have multiple global minima. Specifically, [22] showed that if the training data are linearly separable, then gradient descent on unregularized logistic regression converges directionally to the maximum margin linear classifier on the training data set. [13] further studied the implicit bias of gradient descent for logistic regression on non-separable data. [9] studied the implicit bias of various optimization methods for generic objective functions. [10, 1] established implicit bias results for matrix factorization problems. More recently, [16] showed that gradient flow for learning homogeneous neural networks with logistic loss maximizes the normalized margin on the training data set. These studies of implicit bias offer a handle for us to connect the over-parameterized logistic regression with the maximum margin classifiers for linear models.

## 2 Problem Setting and Notation

**Notations.** We use lower case letters to denote scalars, and use lower/upper case bold face letters to denote vectors/matrices respectively. For a vector $\mathbf{v}$, we denote by $\|\mathbf{v}\|_2$ the $\ell_2$-norm of $\mathbf{v}$. For a matrix $\mathbf{A}$, we use $\|\mathbf{A}\|_2, \|\mathbf{A}\|_F$ to denote its spectral norm and Frobinuous norm respectively, and use $\mathrm{tr}(\mathbf{A})$ to denote its trace. For a vector $\mathbf{v} \in \mathbb{R}^d$ and a positive definite matrix $\mathbf{A}$, we define $\|\mathbf{v}\|_{\mathbf{A}} = \sqrt{\mathbf{v}^\top \mathbf{A} \mathbf{v}}$. For an integer $n$, we denote $[n] = \{1, 2, \ldots, n\}$.

We also use standard asymptotic notations $O(\cdot)$, $\Omega(\cdot)$, $o(\cdot)$, and $\omega(\cdot)$. Let $\{a_n\}$ and $\{b_n\}$ be two sequences. If there exists a constant $C > 0$ such that $|a_n| \leq C|b_n|$ for all large enough $n$, then we denote $a_n = O(b_n)$. We denote $a_n = \Omega(b_n)$ if $b_n = O(a_n)$. Moreover, we write $a_n = o(b_n)$ if $\lim |a_n/b_n| = 0$ and $a_n = \omega(b_n)$ if $\lim |a_n/b_n| = \infty$. We also use $\widetilde{O}(\cdot)$ and $\widetilde{\Omega}(\cdot)$ to hide some logarithmic terms in Big-O and Big-Omega notations.

At last, for a random variable $Z$, we denote by $\|Z\|_{\psi_2}$ and $\|Z\|_{\psi_1}$ the sub-Gaussian and sub-exponential norms of $Z$ respectively.

**Sub-Gaussian Mixture Model.** We consider a model where the feature vectors are generated from a mixture of two sub-Gaussian distributions with means $\boldsymbol{\mu}$ and $-\boldsymbol{\mu}$ and the same covariance matrix $\boldsymbol{\Sigma}$. We assume that each data pair $(\mathbf{x}, y)$ are generated independently from the following procedure:

1. The label $y \in \{+1, -1\}$ is generated as a Rademacher random variable.

2. A random vector $\mathbf{u} \in \mathbb{R}^d$ is generated from a distribution such that the entries of $\mathbf{u}$ are independent sub-Gaussian random variables with $\mathbb{E}[u_j] = 0$, $\mathbb{E}[u_j^2] = 1$ and $\|u_j\|_{\psi_2} \leq \sigma_u$ for all $j \in [d]$.

3. Let $\boldsymbol{\Sigma}$ be a positive definite matrix with eigenvalue decomposition $\boldsymbol{\Sigma} = \mathbf{V}\boldsymbol{\Lambda}\mathbf{V}^\top$, where $\boldsymbol{\Lambda} = \mathrm{diag}\{\lambda_1, \ldots, \lambda_d\}$ and $\mathbf{V}$ is an orthonormal matrix consisting of the eigenvectors of $\boldsymbol{\Sigma}$. We calculate the random vector $\mathbf{q}$ based on $\mathbf{u}$ as $\mathbf{q} = \mathbf{V}\boldsymbol{\Lambda}^{1/2}\mathbf{u}$. This ensures that $\mathbf{q}$ has mean zero and a covariance matrix $\boldsymbol{\Sigma}$.

4. The feature is given as $\mathbf{x} = y \cdot \boldsymbol{\mu} + \mathbf{q}$, where $\boldsymbol{\mu} \in \mathbb{R}^d$ is a vector. Clearly, the mean of $\mathbf{x}$ is $\boldsymbol{\mu}$ when $y = 1$ and is $-\boldsymbol{\mu}$ when $y = -1$.

We consider $n$ training data points $(\mathbf{x}_i, y_i)$ generated independently from the above procedure, and denote

$$\mathbf{X} = \mathbf{y}\boldsymbol{\mu}^\top + \mathbf{Q},$$

where $\mathbf{X} = [\mathbf{x}_1, \ldots, \mathbf{x}_n]^\top, \mathbf{Q} = [\mathbf{q}_1, \ldots, \mathbf{q}_n]^\top \in \mathbb{R}^{n \times d}$, and $\mathbf{y} = [y_1, \ldots, y_n]^\top \in \{\pm 1\}^n$. For any $\boldsymbol{\theta} \in \mathbb{R}^d$, the population risk of the linear classifier $\mathbf{x} \to \langle \boldsymbol{\theta}, \mathbf{x} \rangle$ is defined as:

$$R(\boldsymbol{\theta}) = \mathbb{P}\big(y \cdot \langle \boldsymbol{\theta}, \mathbf{x} \rangle < 0\big).$$

In this paper, we consider the maximum margin linear classifier $\widehat{\boldsymbol{\theta}}_{\text{SVM}}$, i.e., the solution to the hard-margin support vector machine:

$$\widehat{\boldsymbol{\theta}}_{\text{SVM}} = \operatorname{argmin} \|\boldsymbol{\theta}\|_2^2, \text{ subject to } y_i \cdot \langle \boldsymbol{\theta}, \mathbf{x}_i \rangle \geq 1, i \in [n],$$

and study its population risk $R(\widehat{\boldsymbol{\theta}}_{\text{SVM}})$.

A recent work [6] has studied a similar sub-Gaussian mixture model under an assumption that $\operatorname{tr}(\boldsymbol{\Sigma}) = \Omega(d)$, and considered additional label flipping noises. In this paper, we do not introduce the label flipping noises for simplicity, but we consider a general covariance matrix $\boldsymbol{\Sigma}$ to cover the general anisotropic setting. It is worth noting that although our model is not exactly the same as [6] because we don't have additional label flipping noise, there is still noise in our model because of the nature of sub-Gaussian mixture model. For example, consider a mixture of two Gaussian distributions. The two Gaussian clusters have non-trivial overlap, and the Bayes optimal classifier has non-zero Bayes risk as long as $\|\boldsymbol{\mu}\|_2 < \infty$. Therefore, the Bayes optimal classifier and the interpolating classifier are generally quite different. In general, a model is appropriate for the study of benign overfitting whenever the optimal classifier has non-zero Bayes risk.

Our model is rather general and covers the following examples.

**Example 2.1** (Gaussian mixture model). The most straight-forward example is when the data are generated from Gaussian mixtures $N(\boldsymbol{\mu}, \boldsymbol{\Sigma})$ and $N(-\boldsymbol{\mu}, \boldsymbol{\Sigma})$. This is covered by our model when the sub-Gaussian vector $\mathbf{u}$ is a standard Gaussian random vector.

**Example 2.2** (Rare/weak feature model). The rare-weak model is a special case of the Gaussian mixture model where $\boldsymbol{\Sigma} = \mathbf{I}$ and $\boldsymbol{\mu}$ is a sparse vector with $s$ non-zero entries equaling $\gamma$.

The rare/weak feature model was originally investigated by [8, 14], and was recently studied by [6].

**Connection to Over-parameterized Logistic Regression.** Our study of the maximum margin classifier is closely related to over-parameterized logistic regression. In logistic regression, we consider the following empirical loss minimization problem:

$$\min_{\boldsymbol{\theta} \in \mathbb{R}^d} L(\boldsymbol{\theta}) := \frac{1}{n} \sum_{i=1}^{n} \log[1 + \exp(-y_i \cdot \langle \boldsymbol{\theta}, \mathbf{x}_i \rangle)].$$

We solve the above optimization problem with gradient descent

$$\boldsymbol{\theta}^{(t+1)} = \boldsymbol{\theta}^{(t)} - \eta \cdot \nabla L(\boldsymbol{\theta}^{(t)}), \tag{2.1}$$

where $\eta > 0$ is the learning rate.

In the over-parameterized setting where $d \gg n$, it is evident that the training data points are linearly separable with high probability (for example, $\mathbf{X}\mathbf{X}^\top$ is invertible with high probability and the minimum norm interpolator $\widehat{\boldsymbol{\theta}}_{\text{LS}} = \mathbf{X}^\top (\mathbf{X}\mathbf{X}^\top)^{-1}\mathbf{y}$ separates the training data.). For linearly separable data, a series of recent works have studied the *implicit bias* of (stochastic) gradient descent for logistic regression [22, 13, 21]. These results demonstrate that among all linear classifiers that can classify the training data correctly, gradient descent will converge to the one that maximizes the $\ell_2$ margin. Such an implicit bias result is summarized in the following lemma.

**Lemma 2.3** (Theorem 3 in [22]). Suppose that the training data set $\{(\mathbf{x}_i, y_i)\}$ is linearly separable. Then as long as $\eta > 0$ is small enough, the gradient descent iterates $\boldsymbol{\theta}^{(t)}$ for logistic regression defined in (2.1) has the following direction limit:

$$\lim_{t \to \infty} \frac{\boldsymbol{\theta}^{(t)}}{\|\boldsymbol{\theta}^{(t)}\|_2} = \frac{\widehat{\boldsymbol{\theta}}_{\text{SVM}}}{\|\widehat{\boldsymbol{\theta}}_{\text{SVM}}\|_2},$$

where $\widehat{\boldsymbol{\theta}}_{\text{SVM}}$ is the maximum margin classifier.

Lemma 2.3 suggests that our risk bound of the maximum margin classifier $\widehat{\boldsymbol{\theta}}_{\text{SVM}}$ directly implies a risk bound for the over-parameterized logistic regression trained by gradient descent.

# 3 Main Results

In this section, we present our main result on the population risk bound of the maximum margin classifier, and then give a lower bound result to demonstrate the tightness of our upper bounds. We also showcase the application of our results to isotropic and anisotropic sub-Gaussian mixture models to study the conditions under which benign overfitting occurs.

The main result of this paper is given in the following theorem, where we establish the population risk bound for the maximum margin classifier $R(\widehat{\boldsymbol{\theta}}_{\text{SVM}})$.

**Theorem 3.1.** Suppose that $\text{tr}(\boldsymbol{\Sigma}) \geq C \max \left\{ n^{3/2} \|\boldsymbol{\Sigma}\|_2, n\|\boldsymbol{\Sigma}\|_F, n\sqrt{\log(n)} \cdot \|\boldsymbol{\mu}\|_{\boldsymbol{\Sigma}} \right\}$ and $\|\boldsymbol{\mu}\|_2^2 \geq C\|\boldsymbol{\mu}\|_{\boldsymbol{\Sigma}}$ for some absolute constant $C$. Then with probability at least $1 - n^{-1}$, the maximum margin classifier $\widehat{\boldsymbol{\theta}}_{\text{SVM}}$ has the following risk bound

$$R(\widehat{\boldsymbol{\theta}}_{\text{SVM}}) \leq \exp\left( \frac{-C'n\|\boldsymbol{\mu}\|_2^4}{n\|\boldsymbol{\mu}\|_{\boldsymbol{\Sigma}}^2 + \|\boldsymbol{\Sigma}\|_F^2 + n\|\boldsymbol{\Sigma}\|_2^2} \right),$$

where $C'$ is an absolute constant.

Theorem 3.1 gives the population risk bound of the maximum margin classifier $\widehat{\boldsymbol{\theta}}_{\text{SVM}}$. Based on the implicit bias of gradient descent for over-parameterized logistic regression (Lemma 2.3), we have that the gradient descent iterates $\boldsymbol{\theta}^{(t)}$ satisfy that

$$\lim_{t\to\infty} R(\boldsymbol{\theta}^{(t)}) = \lim_{t\to\infty} R(\boldsymbol{\theta}^{(t)}/\|\boldsymbol{\theta}^{(t)}\|_2) = R(\widehat{\boldsymbol{\theta}}_{\text{SVM}}/\|\widehat{\boldsymbol{\theta}}_{\text{SVM}}\|_2) = R(\widehat{\boldsymbol{\theta}}_{\text{SVM}}).$$

Therefore, the same risk bound in Theorem 3.1 also applies to the over-parameterized logistic regression trained by gradient descent.

**Population Risk Lower Bound** We further present lower bounds on the population risk achieved by the maximum margin classifier, which demonstrate that our population risk upper bound in Theorem 3.1 is tight. We have the following theorem.

**Theorem 3.2.** Consider Gaussian mixture model with covariance matrix $\boldsymbol{\Sigma}$ and mean vectors $\boldsymbol{\mu}$ and $-\boldsymbol{\mu}$. Suppose that $\text{tr}(\boldsymbol{\Sigma}) \geq C \max \left\{ n^{3/2} \|\boldsymbol{\Sigma}\|_2, n\|\boldsymbol{\Sigma}\|_F, n\sqrt{\log(n)} \cdot \|\boldsymbol{\mu}\|_{\boldsymbol{\Sigma}} \right\}$, and $\|\boldsymbol{\mu}\|_2^2 \geq C\|\boldsymbol{\mu}\|_{\boldsymbol{\Sigma}}$ for some constant $C$. Then there exist absolute constants $C', C''$, such that the following results hold:

1. If $n\|\boldsymbol{\mu}\|_{\boldsymbol{\Sigma}}^2 \geq C(\|\boldsymbol{\Sigma}\|_F^2 + n\|\boldsymbol{\Sigma}\|_2^2)$, then with probability at least $1 - n^{-1}$,

$$R(\widehat{\boldsymbol{\theta}}_{\text{SVM}}) \geq C'' \exp\left( -C'\|\boldsymbol{\mu}\|_2^4/\|\boldsymbol{\mu}\|_{\boldsymbol{\Sigma}}^2 \right).$$

2. If $\|\boldsymbol{\Sigma}\|_F^2 \geq Cn(\|\boldsymbol{\mu}\|_{\boldsymbol{\Sigma}}^2 + \|\boldsymbol{\Sigma}\|_2^2)$, then with probability at least $1 - n^{-1}$,

$$R(\widehat{\boldsymbol{\theta}}_{\text{SVM}}) \geq C'' \exp\left( -C'n\|\boldsymbol{\mu}\|_2^4/\|\boldsymbol{\Sigma}\|_F^2 \right).$$

Theorem 3.2 gives lower bounds for the population risk in two settings: (i) $n\|\boldsymbol{\mu}\|_{\boldsymbol{\Sigma}}^2 \geq C(\|\boldsymbol{\Sigma}\|_F^2 + n\|\boldsymbol{\Sigma}\|_2^2)$; and (ii) $\|\boldsymbol{\Sigma}\|_F^2 \geq Cn(\|\boldsymbol{\mu}\|_{\boldsymbol{\Sigma}}^2 + \|\boldsymbol{\Sigma}\|_2^2)$. Note that in the population risk upper bound in Theorem 3.1, there are three terms in the denominator of the exponent: $\|\boldsymbol{\mu}\|_{\boldsymbol{\Sigma}}^2$, $\|\boldsymbol{\Sigma}\|_F^2$, and $n\|\boldsymbol{\Sigma}\|_2^2$. Therefore, setting (i) and setting (ii) in Theorem 3.2 correspond to the cases when the first or the second term is the leading term, respectively. Moreover, it is also easy to check that under both settings, our lower bound in Theorem 3.2 matches the upper bound in Theorem 3.1. This suggests that our population risk bound in Theorem 3.1 is tight.

**Implications for Specific Examples.** Theorem 3.1 holds for general covariance matrices $\boldsymbol{\Sigma}$, and illustrates how the spectrum of $\boldsymbol{\Sigma}$ affects the population risk of the maximum margin classifier. This makes our result more general than the recent results in [6, 25], where the population risk bounds are given only in terms of the sample size $n$, dimension $d$ and the norm of the mean vector $\|\boldsymbol{\mu}\|_2$. In fact, when we specialize our general result to the isotropic setting, our result also provides a tighter risk bound than these existing results. Specifically, our population risk bound for the isotropic setting is given in the following corollary.

**Corollary 3.3** (Isotropic sub-Gaussian mixtures). Consider the setting where $\boldsymbol{\Sigma} = \mathbf{I}$. Suppose that $d \geq C \max \left\{ n^2, n\sqrt{\log(n)} \cdot \|\boldsymbol{\mu}\|_2 \right\}$ and $\|\boldsymbol{\mu}\|_2 \geq C$ for some absolute constant $C$. Then with probability at least $1 - n^{-1}$, the maximum margin classifier $\widehat{\boldsymbol{\theta}}_{\text{SVM}}$ has the following risk bound

$$R(\widehat{\boldsymbol{\theta}}_{\text{SVM}}) \leq \exp\left( - \frac{C'n\|\boldsymbol{\mu}\|_2^4}{n\|\boldsymbol{\mu}\|_2^2 + d} \right),$$

where $C'$ is an absolute constant.

**Remark 3.4.** [6] recently gave a risk bound of order $\exp(-\Omega(\|\boldsymbol{\mu}\|_2^4/d))$ for sub-Gaussian mixture models under the condition that $d = \Omega(\max\{n^2 \log(n), n\|\boldsymbol{\mu}\|_2^2\})$. In comparison, our result in Corollary 3.3 only requires the condition $d = \Omega(\max\{n^2, n\sqrt{\log(n)} \cdot \|\boldsymbol{\mu}\|_2\})$, which is milder. Moreover, when the stronger condition $d = \Omega(n\|\boldsymbol{\mu}\|_2^2)$ holds, our risk bound becomes $\exp(-\Omega(n\|\boldsymbol{\mu}\|_2^4/d))$, which is better than the result of [6] by a factor of $n$ in the exponent.

Besides being tighter than previous results when reduced to the isotropic setting, Theorem 3.1 covers both the isotropic and anisotropic settings. In the following, we provide some case studies under the anisotropic setting and show how the decay rate of the eigenvalues of the covariance matrix $\boldsymbol{\Sigma}$ affects the population risk.

It is worth noting that the assumption of Theorem 3.1 requires that $\text{tr}(\boldsymbol{\Sigma})$ is large enough, while the risk bound in Theorem 3.1 only depends on $\|\boldsymbol{\Sigma}\|_F$ and $\|\boldsymbol{\Sigma}\|_2$. In the over-parameterized setting where the dimension $d$ is large, it is possible that for certain covariance matrices $\boldsymbol{\Sigma}$ with appropriate eigenvalue decay rates, $\text{tr}(\boldsymbol{\Sigma}) \gg 1$ while $\|\boldsymbol{\Sigma}\|_F, \|\boldsymbol{\Sigma}\|_2 = O(1)$. This implies that for many anisotropic sub-Gaussian mixture models, the assumptions in Theorem 3.1 can be easily satisfied, while the risk bound can be small at the same time. Following this intuition, we study the conditions under which the the maximum margin interpolator $\widehat{\boldsymbol{\theta}}_{\text{SVM}}$ achieves $o(1)$ population risk. We denote by $\lambda_k$ the $k$-th largest eigenvalue of $\boldsymbol{\Sigma}$, and consider a polynomial decay spectrum $\{\lambda_k = k^{-\alpha}\}_{k=1}^{d}$, where we introduce a parameter $\alpha$ to control the eigenvalue decay rate. We have the following corollary.

**Corollary 3.5** (Anisotripic sub-Gaussian mixtures with polynomial spectrum decay). Suppose that $\lambda_k = k^{-\alpha}$, $n$ is a large enough constant, and one of the following conditions hold:

1. $\alpha \in [0, 1/2)$, $d = \widetilde{\Omega}((\|\boldsymbol{\mu}\|_{\boldsymbol{\Sigma}})^{\frac{1}{1-\alpha}})$, and $\|\boldsymbol{\mu}\|_2 = \omega(1 + d^{1/4 - \alpha/2})$.

2. $\alpha = 1/2$, $d = \widetilde{\Omega}(\|\boldsymbol{\mu}\|_{\boldsymbol{\Sigma}}^2)$, and $\|\boldsymbol{\mu}\|_2 = \omega((\log(d))^{1/4})$.

3. $\alpha \in (1/2, 1)$, $d = \widetilde{\Omega}((\|\boldsymbol{\mu}\|_{\boldsymbol{\Sigma}})^{\frac{1}{1-\alpha}})$, and $\|\boldsymbol{\mu}\|_2 = \omega(1)$.

Then with probability at least $1 - n^{-1}$, the population risk of the maximum margin classifier satisfies $R(\widehat{\boldsymbol{\theta}}_{\text{SVM}}) = o(1)$.

Corollary 3.5 follows by calculating the orders of $\text{tr}(\boldsymbol{\Sigma}) = \sum_{k=1}^{d} \lambda_k$ and $\|\boldsymbol{\Sigma}\|_F^2 = \sum_{k=1}^{d} \lambda_k^2$. Here we treat the sample size $n$ as a constant for simplicity. A full version of the corollary with detailed dependency on $n$ is given as Corollary C.1 in Appendix C together with the proof. Intuitively, when $\|\boldsymbol{\mu}\|_2$ is large, the two classes are far away from each other and therefore linear classifiers can achieve small population risk. From Corollary 3.5, we can see that the decay rate of the eigenvalues of the covariance matrix $\boldsymbol{\Sigma}$ determines how large $\|\boldsymbol{\mu}\|_2$ needs to be to ensure small population risk: when the $\{\lambda_k\}$ decays faster (i.e., when $\alpha$ is larger), the maximum margin classifier can achieve $o(1)$ population risk with a smaller $\|\boldsymbol{\mu}\|_2$.

Corollary 3.5 also exhibits a certain "phase transition" regarding the eigenvalue decay rate and the conditions on $\|\boldsymbol{\mu}\|_2$. We can see that the eigenvalue decay rate can be divided into three regimes $\alpha \in [0, 1/2)$, $\alpha = 1/2$ and $\alpha \in (1/2, 1)$. Under the condition that $d = \widetilde{\Omega}((\|\boldsymbol{\mu}\|_{\boldsymbol{\Sigma}})^{\frac{1}{1-\alpha}})$, achieving $o(1)$ risk in each of these regimes requires $\|\boldsymbol{\mu}\|_2 = \omega(d^{1/4})$, $\|\boldsymbol{\mu}\|_2 = \omega([\log(d)]^{1/4})$, and $\|\boldsymbol{\mu}\|_2 = \omega(1)$ respectively. Specifically, when $\alpha \in (1/2, 1)$, the condition on $\boldsymbol{\mu}$ is independent of the dimension $d$. This means that when $\alpha \in (1/2, 1)$, for any $\epsilon > 0$, as long as $\|\boldsymbol{\mu}\|_2 = \Omega(\sqrt{\log(\epsilon)})$, we have

$$\lim_{d \to \infty} R(\widehat{\boldsymbol{\theta}}_{\text{SVM}}) \leq \epsilon.$$

Therefore, our result covers the infinite dimensional setting when the eigenvalues of the covariance matrix $\mathbf{\Sigma}$ have an appropriate decay rate, i.e., $\alpha \in (1/2, 1)$.

Another interesting observation in Theorem 3.1 is that it uses both $\|\boldsymbol{\mu}\|_{\mathbf{\Sigma}}$ and $\|\boldsymbol{\mu}\|_2$, and therefore the alignment between $\boldsymbol{\mu}$ and the eigenvectors of $\mathbf{\Sigma}$ can affect the population risk bound. In our discussion above, we have mainly focused on the worst case scenario where the direction of $\boldsymbol{\mu}$ aligns with the first eigenvector of $\mathbf{\Sigma}$. In the following corollary, we discuss the case where $\boldsymbol{\mu}$ is parallel to the eigenvector of $\mathbf{\Sigma}$ corresponding to the eigenvalue $\lambda_k$.

**Corollary 3.6** (Risk bounds for $\boldsymbol{\mu}$ along different directions). Suppose that $\mathbf{\Sigma}\boldsymbol{\mu} = \lambda_k \boldsymbol{\mu}$ for some $k \in [d]$, $\mathrm{tr}(\mathbf{\Sigma}) \geq C \max\left\{n^{3/2}\|\mathbf{\Sigma}\|_2, n\|\mathbf{\Sigma}\|_F, n\sqrt{\lambda_k \log(n)} \cdot \|\boldsymbol{\mu}\|_2\right\}$ and $\|\boldsymbol{\mu}\|_2^2 \geq C\lambda_k$ for some absolute constant $C$. Then with probability at least $1 - n^{-1}$, the maximum margin classifier $\widehat{\boldsymbol{\theta}}_{\mathrm{SVM}}$ has the following risk bound

$$R(\widehat{\boldsymbol{\theta}}_{\mathrm{SVM}}) \leq \exp\left(\frac{-C'n\|\boldsymbol{\mu}\|_2^4}{n\lambda_k \cdot \|\boldsymbol{\mu}\|_2^2 + \|\mathbf{\Sigma}\|_F^2 + n\|\mathbf{\Sigma}\|_2^2}\right),$$

where $C'$ is an absolute constant.

We can see that, when $\boldsymbol{\mu}$ aligns with the eigendirections corresponding to a smaller eigenvalue of $\mathbf{\Sigma}$, then Corollary 3.6 holds under milder conditions on $\mathrm{tr}(\mathbf{\Sigma})$ and $\|\boldsymbol{\mu}\|_2$, and the population risk achieved by the maximum margin solution is also better. This phenomenon perfectly matches the geometric intuition of sub-Gaussian mixture classifications, as is illustrated in Figure 1.

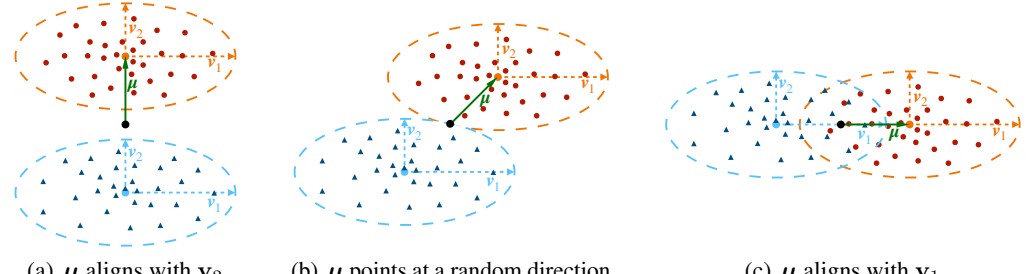

(a) $\boldsymbol{\mu}$ aligns with $\mathbf{v}_2$     (b) $\boldsymbol{\mu}$ points at a random direction     (c) $\boldsymbol{\mu}$ aligns with $\mathbf{v}_1$

Figure 1: A 2-dimensional illustration of sub-Gausisan mixture classification problems with different directions of $\boldsymbol{\mu}$. We consider the setting where $\mathbf{\Sigma} \in \mathbb{R}^{2 \times 2}$ has two eigenvalues $\lambda_1 > \lambda_2$ with the corresponding eigenvectors $\mathbf{v}_1, \mathbf{v}_2$. (a) shows the setting where $\boldsymbol{\mu}$ aligns with $\mathbf{v}_2$. (b) shows the setting where $\boldsymbol{\mu}$ points at a random direction. (c) is for the case when $\boldsymbol{\mu}$ aligns with $\mathbf{v}_1$. These figures clearly show that (a) is the easiest case for classification and (c) is the hardest case. This matches the result in Corollary 3.6.

At last, we can also apply our risk bound to the rare/weak feature model defined in Example 2.2. We have the following corollary.

**Corollary 3.7** (Rare/weak feature model). Consider the rare/weak feature model (Example 2.2). Suppose that $d \geq C \max\{n^2, \gamma n\sqrt{s \log(n)}\}$ and $\gamma\sqrt{s} \geq C$ for some large enough absolute constant $C$. Then when $n$ is large enough, with probability at least $1 - n^{-1}$, the maximum margin classifier $\widehat{\boldsymbol{\theta}}_{\mathrm{SVM}}$ has the following risk bound

$$R(\widehat{\boldsymbol{\theta}}_{\mathrm{SVM}}) \leq \exp\left(-\frac{C'n\gamma^4 s^2}{n\gamma^2 s + d}\right),$$

where $C'$ is an absolute constant.

By Corollary 3.7, we can see that our bound is tighter by a factor of $n$ in the exponent compared with the risk bound in [6] for the rare/weak feature model. Under the setting where $n$ and $\gamma$ are fixed constants, our bound can also be compared with the negative result in [14], which showed that achieving a small population risk is impossible when $s = O(d^2)$. Our result, on the other hand, demonstrates that when $s = \omega(d^2)$, $o(1)$ population risk is achievable.

# 4 Proof of the Main Results

In this section, we explain how we establish the population risk bound of the maximum margin classifier, and give the proof of Theorem 3.1.

For classification problems, one of the key challenges is that the maximum margin classifier usually does not have an explicit form solution. To overcome this difficulty, [6] utilized the implicit bias results (Lemma 2.3) to get a handle on the relationship between the maximum margin classifier and the training data. More recently, [25] showed that for isotropic Gaussian mixture models, an explicit form of $\widehat{\boldsymbol{\theta}}_{\text{SVM}}$ can be calculated by the equivalence between hard-margin support vector machine and minimum norm least square regression. Notably, it was shown that such an equivalence result holds under the assumptions of [6] and no any additional assumptions are needed. In this paper, we also study the equivalence between classification and regression as a first step. However, our proof works for a more general setting that covers both isotropic and anisotropic sub-Gaussian mixtures, and introduces a novel proof technique based on the polarization identity that leads to a tighter bound. We present this result in Section 4.

**Step 1. Equivalence Between Classification and Regression.** Here we establish an equivalence guarantee for the maximum margin classifier and the minimum norm interpolator. Note that the definitions of the minimum norm interpolator $\widehat{\boldsymbol{\theta}}_{\text{LS}}$ and the maximum margin classifier $\widehat{\boldsymbol{\theta}}_{\text{SVM}}$ are as follows:

$$\widehat{\boldsymbol{\theta}}_{\text{LS}} := \operatorname{argmin} \|\boldsymbol{\theta}\|_2^2, \;\; \text{subject to } y_i \cdot \langle \boldsymbol{\theta}, \mathbf{x}_i \rangle = 1, i \in [n],$$

$$\widehat{\boldsymbol{\theta}}_{\text{SVM}} = \operatorname{argmin} \|\boldsymbol{\theta}\|_2^2, \;\; \text{subject to } y_i \cdot \langle \boldsymbol{\theta}, \mathbf{x}_i \rangle \geq 1, i \in [n].$$

We can see that the two optimization problems have the same solution when all the training data are support vectors, i.e., all the inequalities become equalities in the constraints [19, 12]. Here we derive such an equivalence result for sub-Gaussian mixture models. The result is as follows.

**Proposition 4.1.** Suppose that $\operatorname{tr}(\boldsymbol{\Sigma}) \geq C \max\{n^{3/2}\|\boldsymbol{\Sigma}\|_2, n\|\boldsymbol{\Sigma}\|_F, n\sqrt{\log(n)} \cdot \|\boldsymbol{\mu}\|_{\boldsymbol{\Sigma}}\}$ for some absolute constant $C$. Then with probability at least $1 - O(n^{-2})$, $\widehat{\boldsymbol{\theta}}_{\text{SVM}} = \widehat{\boldsymbol{\theta}}_{\text{LS}}$.

The proof of Proposition 4.1 utilizes an argument based on the polarization identity to give a tight bound, which may be of independent interest. The details are given in Appendix 4.1.

**Step 2. Population Risk of the Maximum Margin Classifier.** We derive the population risk bound for the maximum margin classifier and provide the proof of Theorem 3.1. We first present the following lemma on the risk bound of linear classifiers for sub-Gaussian mixture models.

**Lemma 4.2.** There exists an absolute constant $C$ such that, for any $\boldsymbol{\theta} \in \mathbb{R}^d$, the following risk bound holds:

$$R(\boldsymbol{\theta}) \leq \exp\left( -\frac{C(\boldsymbol{\theta}^\top \boldsymbol{\mu})^2}{\|\boldsymbol{\theta}\|_{\boldsymbol{\Sigma}}^2} \right).$$

A similar result is given in [6] where $\boldsymbol{\Sigma}$ is replaced by $\mathbf{I}$. Our result here depends on the full spectrum of the covariance matrix and is sharper than [6] when $\boldsymbol{\Sigma}$ has decaying eigenvalues.

The proof of Lemma 4.2 is given in Appendix A.2. In addition to this risk bound for general vector $\boldsymbol{\theta}$, we also have the following explicit calculation for $\widehat{\boldsymbol{\theta}}_{\text{SVM}}$ thanks to our analysis in Section 4. This is because the minimum norm interpolator $\widehat{\boldsymbol{\theta}}_{\text{LS}}$ has the explicit form $\widehat{\boldsymbol{\theta}}_{\text{LS}} = \mathbf{X}^\top(\mathbf{X}\mathbf{X}^\top)^{-1}\mathbf{y}$. Therefore by Proposition 4.1, we also have $\widehat{\boldsymbol{\theta}}_{\text{SVM}} = \mathbf{X}^\top(\mathbf{X}\mathbf{X}^\top)^{-1}\mathbf{y}$. Plugging this calculation into the risk bound in Lemma 4.2 and utilizing the model definition $\mathbf{X} = \mathbf{y}\boldsymbol{\mu}^\top + \mathbf{Q}$, we are able to show the following risk bound for $\widehat{\boldsymbol{\theta}}_{\text{SVM}}$.

**Lemma 4.3.** Suppose that $\operatorname{tr}(\boldsymbol{\Sigma}) \geq C \max\{n\sqrt{\log(n)}, n^{3/2}\|\boldsymbol{\Sigma}\|_2, n\|\boldsymbol{\Sigma}\|_F, n\|\boldsymbol{\mu}\|_{\boldsymbol{\Sigma}}\}$ for some absolute constant $C$. Then with probability at least $1 - O(n^{-2})$,

$$R(\widehat{\boldsymbol{\theta}}_{\text{SVM}}) \leq \exp\left\{ \frac{-C' \cdot [\mathbf{y}^\top(\mathbf{X}\mathbf{X}^\top)^{-1}\mathbf{X}\boldsymbol{\mu}]^2}{(\mathbf{y}^\top(\mathbf{X}\mathbf{X}^\top)^{-1}\mathbf{y})^2\|\boldsymbol{\mu}\|_{\boldsymbol{\Sigma}}^2 + \|\mathbf{Q}^\top(\mathbf{X}\mathbf{X}^\top)^{-1}\mathbf{y}\|_{\boldsymbol{\Sigma}}^2} \right\},$$

where $C'$ is an absolute constant.

Lemma 4.3 utilizes the structure of the model to divide the denominator in the exponent into two terms. Motivated by this result, we define

$$I_1 = [\mathbf{y}^\top (\mathbf{XX}^\top)^{-1}\mathbf{X}\boldsymbol{\mu}]^2, \;\; I_2 = (\mathbf{y}^\top (\mathbf{XX}^\top)^{-1}\mathbf{y})^2 \cdot \|\boldsymbol{\mu}\|_{\boldsymbol{\Sigma}}^2, \;\; I_3 = \|\mathbf{Q}^\top(\mathbf{XX}^\top)^{-1}\mathbf{y}\|_{\boldsymbol{\Sigma}}^2.$$

This leads to our analysis in the next step.

**Step 3. Bounds for $I_1$, $I_2$ and $I_3$.** In the following, we develop a lower bound for $I_1$ and upper bounds for $I_2$ and $I_3$ respectively. The following lemma summarizes the bounds.

**Lemma 4.4.** Suppose that $\mathrm{tr}(\boldsymbol{\Sigma}) \geq C \max\{n, n\|\boldsymbol{\Sigma}\|_2, \sqrt{n}\|\boldsymbol{\Sigma}\|_F, n\|\boldsymbol{\mu}\|_{\boldsymbol{\Sigma}}\}$ and $\|\boldsymbol{\mu}\|_2^2 \geq C\|\boldsymbol{\mu}\|_{\boldsymbol{\Sigma}}$ for some absolute constant $C$. Then when $n$ is large enough, with probability at least $1 - O(n^{-2})$,

$$I_1 \geq C'^{-1} H(\boldsymbol{\mu}, \mathbf{Q}, \mathbf{y}, \boldsymbol{\Sigma}) \cdot n^2 \cdot \|\boldsymbol{\mu}\|_2^4,$$
$$I_2 \leq C' H(\boldsymbol{\mu}, \mathbf{Q}, \mathbf{y}, \boldsymbol{\Sigma}) \cdot n^2 \cdot \|\boldsymbol{\mu}\|_{\boldsymbol{\Sigma}}^2,$$
$$I_3 \leq C' H(\boldsymbol{\mu}, \mathbf{Q}, \mathbf{y}, \boldsymbol{\Sigma}) \cdot (n \cdot \|\boldsymbol{\Sigma}\|_F^2 + n^2 \cdot \|\boldsymbol{\Sigma}\|_2^2),$$

where $H(\boldsymbol{\mu}, \mathbf{Q}, \mathbf{y}, \boldsymbol{\Sigma}) > 0$ is a strictly positive coefficient, and $C' > 0$ is an absolute constant.

The proof of Lemma 4.4 is given in Appendix A.2. To illustrate the key idea in the proof of Lemma 4.4, we take $I_3$ as an example. Based on our model in Section 2, we have $\mathbf{Q} = \mathbf{Z}\boldsymbol{\Lambda}^{1/2}\mathbf{V}^\top$, where $\mathbf{Z} \in \mathbb{R}^{n \times d}$ is a random matrix with independent sub-Gaussian entries, and $\boldsymbol{\Lambda}, \mathbf{V}$ are defined based on the eigenvalue decomposition $\boldsymbol{\Sigma} = \mathbf{V}\boldsymbol{\Lambda}\mathbf{V}^\top$. By some linear algebra calculation (see the proof of Lemma 4.4 in Appendix A.2 for more details), we have

$$I_3 = \mathbf{a}^\top (\mathbf{Z}\boldsymbol{\Lambda}\mathbf{Z}^\top)^{-1}\mathbf{Z}\boldsymbol{\Lambda}^2\mathbf{Z}^\top(\mathbf{Z}\boldsymbol{\Lambda}\mathbf{Z}^\top)^{-1}\mathbf{a}, \tag{4.1}$$

where $\|\mathbf{a}\|_2^2 = O(D^{-2}n)$ with

$$D = \mathbf{y}^\top (\mathbf{QQ}^\top)^{-1}\mathbf{y} \cdot (\|\boldsymbol{\mu}\|_2^2 - \boldsymbol{\mu}^\top \mathbf{Q}^\top(\mathbf{QQ}^\top)^{-1}\mathbf{Q}\boldsymbol{\mu}) + (1 + \mathbf{y}^\top(\mathbf{QQ}^\top)^{-1}\mathbf{Q}\boldsymbol{\mu})^2.$$

The key observation here is that while the term $D$ above has a very complicated form, it is not necessary to bound it. This is because $D^{-2}$ is a common term that appears in all $I_1$, $I_2$ and $I_3$ and therefore can be canceled out when calculating the ratio $I_1/(I_2 + I_3)$. With the calculation in (4.1), we are able to invoke the following eigenvalue concentration inequalities (see Lemma A.4 and Lemma A.7 for more details) to give upper and lower bounds regarding the matrices $\mathbf{Z}\boldsymbol{\Lambda}^2\mathbf{Z}^\top$ and $\mathbf{Z}\boldsymbol{\Lambda}\mathbf{Z}^\top$ respectively:

$$\left\|\mathbf{Z}\boldsymbol{\Lambda}\mathbf{Z}^\top - \mathrm{tr}(\boldsymbol{\Sigma})\cdot\mathbf{I}\right\|_2 \leq c_1 \cdot \left(n\|\boldsymbol{\Sigma}\|_2 + \sqrt{n}\|\boldsymbol{\Sigma}\|_F\right),$$
$$\left\|\mathbf{Z}\boldsymbol{\Lambda}^2\mathbf{Z}^\top - \|\boldsymbol{\Sigma}\|_F^2\cdot\mathbf{I}\right\|_2 \leq c_1 \cdot \left(n\|\boldsymbol{\Sigma}\|_2^2 + \sqrt{n}\|\boldsymbol{\Sigma}^2\|_F\right),$$

where $c_1$ is an absolute constant. Plugging the above inequalities and the bound $\|\mathbf{a}\|_2^2 = O(D^{-2}n)$ into (4.1), we obtain with some calculation that

$$I_3 \leq c_2 H(\boldsymbol{\mu}, \mathbf{Q}, \mathbf{y}, \boldsymbol{\Sigma}) \cdot (n \cdot \|\boldsymbol{\Sigma}\|_F^2 + n^2 \cdot \|\boldsymbol{\Sigma}\|_2^2)$$

with $H(\boldsymbol{\mu}, \mathbf{Q}, \mathbf{y}, \boldsymbol{\Sigma}) = [D \cdot \mathrm{tr}(\boldsymbol{\Sigma})]^{-2}$, where $c_2$ is an absolute constant. This gives the bound of $I_3$.

Lemma 4.4 is significant in three-fold. First of all, the result does not have an explicit dependency on $d$, which makes it applicable to infinite dimensional data. Second, Lemma 4.4 gives bounds with great simplicity, and shows that the three bounds share a same strictly positive factor $H(\boldsymbol{\mu}, \mathbf{Q}, \mathbf{y}, \boldsymbol{\Sigma})$, which can be canceled out since our final goal is to bound the ratio $I_1/(I_2 + I_3)$. Lastly, Lemma 4.4 reveals the fact that the risk bound only depends on $\|\boldsymbol{\Sigma}\|_F$ and $\|\boldsymbol{\Sigma}\|_2$, which can be small even though the assumption requires $\mathrm{tr}(\boldsymbol{\Sigma})$ to be large.

We are now ready to present the proof of Theorem 3.1.

*Proof of Theorem 3.1.* Clearly, under the assumptions of Theorem 3.1, the conditions in Lemma 4.3 and Lemma 4.4 are both satisfied. By Lemma 4.4, we have

$$\frac{[\mathbf{y}^\top(\mathbf{XX}^\top)^{-1}\mathbf{X}\boldsymbol{\mu}]^2}{(\mathbf{y}^\top(\mathbf{XX}^\top)^{-1}\mathbf{y})^2\|\boldsymbol{\mu}\|_{\boldsymbol{\Sigma}}^2 + \|\mathbf{Q}^\top(\mathbf{XX}^\top)^{-1}\mathbf{y}\|_{\boldsymbol{\Sigma}}^2} \geq c_1 \cdot \frac{n^2\|\boldsymbol{\mu}\|_2^4}{n^2\|\boldsymbol{\mu}\|_{\boldsymbol{\Sigma}}^2 + n\cdot\|\boldsymbol{\Sigma}\|_F^2 + n^2\cdot\|\boldsymbol{\Sigma}\|_2^2},$$

where $c_1$ is an absolute constant. Therefore by Lemma 4.3 we have

$$R(\widehat{\boldsymbol{\theta}}_{\mathrm{SVM}}) \leq \exp\left(\frac{-c_2 n\|\boldsymbol{\mu}\|_2^4}{n\|\boldsymbol{\mu}\|_{\boldsymbol{\Sigma}}^2 + \|\boldsymbol{\Sigma}\|_F^2 + n\|\boldsymbol{\Sigma}\|_2^2}\right)$$

for some absolute constant $c_2$. Note that by union bound, the above inequality holds with probability at least $1 - O(n^{-2}) \geq 1 - n^{-1}$ when $n$ is large enough. This completes the proof. $\qquad\square$

## 5 Conclusion and Future Work

We have studied the benign overfitting phenomenon for sub-Gaussian mxiture models, and established a population risk bound for the maximum margin classifier. Our population risk bound is general and covers both the isotropic and anisotropic settings. When reduced to the isotropic setting, our bound is tighter than existing results. We have also studied a class of non-isotropic models which can be benign even for infinite-dimensional data.

An interesting future work direction is to study the relation between the dimension and the population risk and verify the double descent phenomenon. Studying benign overfitting for more complicated learning models such as neural networks is another important future work direction.

## Acknowledgments and Disclosure of Funding

We thank the anonymous reviewers for their helpful comments. This work was done when YC was a postdoctoral researcher at UCLA. YC and QG are partially supported by the National Science Foundation award IIS-1903202 and IIS-2008981. MB acknowledges support from NSF IIS-1815697 and the support of the NSF and the Simons Foundation for the Collaboration on the Theoretical Foundations of Deep Learning through awards DMS-2031883 and #814639. The views and conclusions contained in this paper are those of the authors and should not be interpreted as representing any funding agencies.

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
