# A  Completing the Proof of Theorem 3.1

In Section 4 we give the proof of Theorem 3.1 based on several technical propositions and lemmas. Here we present the complete proof of these propositions and lemmas.

## A.1  Proof of Proposition 4.1

Here we present the proof of Proposition 4.1. We begin with the following lemma, which is studied by [19, 12].

**Lemma A.1** ([12]). $\widehat{\boldsymbol{\theta}}_{\text{SVM}} = \widehat{\boldsymbol{\theta}}_{\text{LS}}$ if and only if $\mathbf{y}^\top(\mathbf{XX}^\top)^{-1}\mathbf{e}_i y_i > 0$ for all $i \in [n]$.

According to Lemma A.1, to study the equivalence between the maximum margin classifier and the minimum norm interpolator, it suffices to derive sufficient conditions such that $\mathbf{y}^\top(\mathbf{XX}^\top)^{-1}\mathbf{e}_i y_i$, $i \in [n]$ are strictly positive with high probability. We have the following lemma which summarizes some calculations regarding the quantity $\mathbf{y}^\top(\mathbf{XX}^\top)^{-1}\mathbf{e}_i y_i$.

**Lemma A.2.** Suppose that $\text{tr}(\boldsymbol{\Sigma}) > C \max\{n^{3/2}\|\boldsymbol{\Sigma}\|_2, n\|\boldsymbol{\Sigma}\|_F, n\|\boldsymbol{\mu}\|_{\boldsymbol{\Sigma}}\}$ for some absolute constant $C$. Then with probability at least $1 - O(n^{-2})$,

$$\mathbf{y}^\top(\mathbf{XX}^\top)^{-1}\mathbf{e}_i y_i \geq G\Big[1 - C'n\big|\boldsymbol{\mu}^\top\mathbf{Q}^\top(\mathbf{QQ}^\top)^{-1}\mathbf{e}_i\big|\Big]$$

for all $i \in [n]$, where $G = G(\boldsymbol{\mu}, \mathbf{Q}, \mathbf{y}, \boldsymbol{\Sigma}) > 0$ is a strictly positive factor and $C' > 0$ is an absolute constant.

By Lemma A.2, we can see that in order to ensure $\mathbf{y}^\top(\mathbf{XX}^\top)^{-1}\mathbf{e}_i y_i > 0$, it suffices to establish an upper bound for $|\boldsymbol{\mu}^\top\mathbf{Q}^\top(\mathbf{QQ}^\top)^{-1}\mathbf{e}_i|$. However, deriving tight upper bounds for this term turns out to be challenging, as a simple application of the Cauchy-Schwarz inequality can lead to a loose bound with an additional $\sqrt{n}$ factor. In the following, we establish a refined bound on the term $|\boldsymbol{\mu}^\top\mathbf{Q}^\top(\mathbf{QQ}^\top)^{-1}\mathbf{e}_i|$.

**Lemma A.3.** Suppose that $\text{tr}(\boldsymbol{\Sigma}) > C \max\{n^{3/2}\|\boldsymbol{\Sigma}\|_2, n\|\boldsymbol{\Sigma}\|_F\}$ for some absolute constant $C$. Then with probability at least $1 - O(n^{-2})$,

$$\big|\boldsymbol{\mu}^\top\mathbf{Q}^\top(\mathbf{QQ}^\top)^{-1}\mathbf{e}_i\big| \leq \frac{C'\|\boldsymbol{\mu}\|_{\boldsymbol{\Sigma}} \cdot \sqrt{\log(n)}}{\text{tr}(\boldsymbol{\Sigma})}$$

for all $i \in [n]$, where $C' > 0$ is an absolute constant.

We are now ready to present the proof of Proposition 4.1

*Proof of Proposition 4.1.* By the union bound, we have that with probability at least $1 - 2n^{-2}$, the results in Lemma A.2 and Lemma A.3 both hold. Therefore, for any $i \in [n]$, we have

$$\mathbf{y}^\top(\mathbf{XX}^\top)^{-1}\mathbf{e}_i y_i \geq G\Big[1 - c_1 n\big|\boldsymbol{\mu}^\top\mathbf{Q}^\top(\mathbf{QQ}^\top)^{-1}\mathbf{e}_i\big|\Big] \geq G\left[1 - \frac{c_2 n\sqrt{\log(n)} \cdot \|\boldsymbol{\mu}\|_{\boldsymbol{\Sigma}}}{\text{tr}(\boldsymbol{\Sigma})}\right]$$

$$\propto \text{tr}(\boldsymbol{\Sigma}) - c_2 n\sqrt{\log(n)} \cdot \|\boldsymbol{\mu}\|_{\boldsymbol{\Sigma}}.$$

By the assumption $\text{tr}(\boldsymbol{\Sigma}) \geq Cn\sqrt{\log(n)} \cdot \|\boldsymbol{\mu}\|_{\boldsymbol{\Sigma}}$ for some large enough absolute constant $C$, we have $\mathbf{y}^\top(\mathbf{XX}^\top)^{-1}\mathbf{e}_i y_i > 0$. Finally, applying Lemma A.1, we conclude that $\widehat{\boldsymbol{\theta}}_{\text{SVM}} = \widehat{\boldsymbol{\theta}}_{\text{LS}}$. □

## A.2  Proof of Lemmas in Section 4

We denote $\boldsymbol{\nu} = \mathbf{Q}\boldsymbol{\mu}$ and $\mathbf{A} = \mathbf{QQ}^\top$. Based on these notations, in the following we present several basic lemmas that are used in our proof. We have the following lemma which gives concentration inequalities for the the eigenvalues of $\mathbf{A}$.

**Lemma A.4.** With probability at least $1 - n^{-2}$,

$$\big\|\mathbf{A} - \text{tr}(\boldsymbol{\Sigma}) \cdot \mathbf{I}\big\|_2 \leq \epsilon_\lambda := C\sigma_u^2\big(n \cdot \|\boldsymbol{\Sigma}\|_2 + \sqrt{n} \cdot \|\boldsymbol{\Sigma}\|_F\big),$$

where $C$ is an absolute constant.

The following lemma presents some calculations on the quantity $\mathbf{y}^\top(\mathbf{X}\mathbf{X}^\top)^{-1}$. It utilizes a result introduced in [25], which is based on the application of the Sherman–Morrison–Woodbury formula.

**Lemma A.5.** The following calculation of $\mathbf{y}^\top(\mathbf{X}\mathbf{X}^\top)^{-1}$ holds:

$$\mathbf{y}^\top(\mathbf{X}\mathbf{X}^\top)^{-1} = D^{-1}[(1 + \mathbf{y}^\top\mathbf{A}^{-1}\boldsymbol{\nu}) \cdot \mathbf{y}^\top\mathbf{A}^{-1} - \mathbf{y}^\top\mathbf{A}^{-1}\mathbf{y} \cdot \boldsymbol{\nu}^\top\mathbf{A}^{-1}],$$

where $D = \mathbf{y}^\top\mathbf{A}^{-1}\mathbf{y} \cdot (\|\boldsymbol{\mu}\|_2^2 - \boldsymbol{\nu}^\top\mathbf{A}^{-1}\boldsymbol{\nu}) + (1 + \mathbf{y}^\top\mathbf{A}^{-1}\boldsymbol{\nu})^2 > 0$.

Motivated by Lemma A.5, we estimate the orders of the terms $\mathbf{y}^\top\mathbf{A}^{-1}\mathbf{y}$, $\boldsymbol{\nu}^\top\mathbf{A}^{-1}\boldsymbol{\nu}$, and $\mathbf{y}^\top\mathbf{A}^{-1}\boldsymbol{\nu}$. The results are given in the following lemma.

**Lemma A.6.** Let $\epsilon_\lambda$ be defined in Lemma A.4, and suppose that $\mathrm{tr}(\boldsymbol{\Sigma}) > \epsilon_\lambda$. Then with probability at least $1 - O(n^{-2})$, the following inequalities hold:

$$\frac{n}{\mathrm{tr}(\boldsymbol{\Sigma}) + \epsilon_\lambda} \le \mathbf{y}^\top\mathbf{A}^{-1}\mathbf{y} \le \frac{n}{\mathrm{tr}(\boldsymbol{\Sigma}) - \epsilon_\lambda},$$

$$\frac{n - C\sqrt{n\log(n)}}{\mathrm{tr}(\boldsymbol{\Sigma}) + \epsilon_\lambda} \cdot \|\boldsymbol{\mu}\|_{\boldsymbol{\Sigma}}^2 \le \boldsymbol{\nu}^\top\mathbf{A}^{-1}\boldsymbol{\nu} \le \frac{n + C\sqrt{n\log(n)}}{\mathrm{tr}(\boldsymbol{\Sigma}) - \epsilon_\lambda} \cdot \|\boldsymbol{\mu}\|_{\boldsymbol{\Sigma}}^2,$$

$$|\mathbf{y}^\top\mathbf{A}^{-1}\boldsymbol{\nu}| \le \frac{Cn}{\mathrm{tr}(\boldsymbol{\Sigma}) - \epsilon_\lambda}\|\boldsymbol{\mu}\|_{\boldsymbol{\Sigma}},$$

where $C$ is an absolute constant.

### A.2.1  Proof of Lemma 4.2

Here we give the detailed proof of Lemma 4.2, which is based on the one-side sub-Gaussian tail bound.

*Proof of Lemma 4.2.* By definition, we have

$$R(\boldsymbol{\theta}) = \mathbb{P}(y \cdot \boldsymbol{\theta}^\top\mathbf{x} < 0) = \mathbb{P}[y \cdot \boldsymbol{\theta}^\top(y \cdot \boldsymbol{\mu} + \mathbf{q}) < 0] = \mathbb{P}[\boldsymbol{\theta}^\top\boldsymbol{\mu} < y \cdot \boldsymbol{\theta}^\top\mathbf{q}] = \mathbb{P}[\boldsymbol{\theta}^\top\boldsymbol{\mu} < y \cdot \boldsymbol{\theta}^\top\mathbf{V}\boldsymbol{\Lambda}^{1/2}\mathbf{u}],$$

where in the second and last equations we plug in the definitions of $\mathbf{x}$ and $\mathbf{q}$ according to our data generation procedure described in Section 2. Note that $\mathbf{u}$ has independent, $\sigma_u$-sub-Gaussian entries. Therefore we have

$$\|\boldsymbol{\theta}^\top\mathbf{V}\boldsymbol{\Lambda}^{1/2}\mathbf{u}\|_{\psi_2} \le c_1\|\boldsymbol{\theta}^\top\mathbf{V}\boldsymbol{\Lambda}^{1/2}\|_2 = c_1\sqrt{\boldsymbol{\theta}^\top\mathbf{V}\boldsymbol{\Lambda}\mathbf{V}^\top\boldsymbol{\theta}} = c_1\sqrt{\boldsymbol{\theta}^\top\boldsymbol{\Sigma}\boldsymbol{\theta}}.$$

Applying the one-side sub-Gaussian tail bound (e.g., Theorem A.2 in [6]) completes the proof. □

### A.2.2  Proof of Lemma 4.3

The proof of Lemma 4.3 is given as follows, where we utilize Proposition 4.1 and Lemma 4.2 to derive the desired bound.

*Proof of Lemma 4.3.* By Proposition 4.1, we have

$$\widehat{\boldsymbol{\theta}}_{\mathrm{SVM}} = \widehat{\boldsymbol{\theta}}_{\mathrm{LS}} = \mathbf{X}^\top(\mathbf{X}\mathbf{X}^\top)^{-1}\mathbf{y}.$$

Plugging it into the risk bound in Lemma 4.2, we obtain

$$R(\widehat{\boldsymbol{\theta}}_{\mathrm{SVM}}) \le \exp\left\{ -\frac{C[\mathbf{y}^\top(\mathbf{X}\mathbf{X}^\top)^{-1}\mathbf{X}\boldsymbol{\mu}]^2}{\|\mathbf{X}^\top(\mathbf{X}\mathbf{X}^\top)^{-1}\mathbf{y}\|_{\boldsymbol{\Sigma}}^2} \right\}.$$

Note that based on our model, we have $\mathbf{X} = \mathbf{y}\boldsymbol{\mu}^\top + \mathbf{Q}$, and

$$\begin{aligned}
\|\mathbf{X}^\top(\mathbf{X}\mathbf{X}^\top)^{-1}\mathbf{y}\|_{\boldsymbol{\Sigma}}^2 &= \|(\mathbf{y}\boldsymbol{\mu}^\top + \mathbf{Q})^\top(\mathbf{X}\mathbf{X}^\top)^{-1}\mathbf{y}\|_{\boldsymbol{\Sigma}}^2 \\
&\le 2\|\boldsymbol{\mu}\mathbf{y}^\top(\mathbf{X}\mathbf{X}^\top)^{-1}\mathbf{y}\|_{\boldsymbol{\Sigma}}^2 + 2\|\mathbf{Q}^\top(\mathbf{X}\mathbf{X}^\top)^{-1}\mathbf{y}\|_{\boldsymbol{\Sigma}}^2 \\
&= 2(\mathbf{y}^\top(\mathbf{X}\mathbf{X}^\top)^{-1}\mathbf{y})^2 \cdot \|\boldsymbol{\mu}\|_{\boldsymbol{\Sigma}}^2 + 2\|\mathbf{Q}^\top(\mathbf{X}\mathbf{X}^\top)^{-1}\mathbf{y}\|_{\boldsymbol{\Sigma}}^2.
\end{aligned}$$

Therefore we have

$$R(\widehat{\boldsymbol{\theta}}_{\mathrm{SVM}}) \le \exp\left\{ \frac{-(C/2) \cdot [\mathbf{y}^\top(\mathbf{X}\mathbf{X}^\top)^{-1}\mathbf{X}\boldsymbol{\mu}]^2}{(\mathbf{y}^\top(\mathbf{X}\mathbf{X}^\top)^{-1}\mathbf{y})^2 \cdot \|\boldsymbol{\mu}\|_{\boldsymbol{\Sigma}}^2 + \|\mathbf{Q}^\top(\mathbf{X}\mathbf{X}^\top)^{-1}\mathbf{y}\|_{\boldsymbol{\Sigma}}^2} \right\}.$$

This completes the proof. □

### A.2.3 Proof of Lemma 4.4

In this subsection we present the proof of Lemma 4.4. We first give the following lemma, which follows by exactly the same proof as Lemma A.4.

**Lemma A.7.** Suppose that $\mathbf{Z} \in \mathbb{R}^{n \times d}$ is a random matrix with i.i.d. sub-Gaussian entries with sub-Gaussian norm $\sigma_u$. Then with probability at least $1 - O(n^{-2})$,

$$\left\| \mathbf{Z}\boldsymbol{\Lambda}^2\mathbf{Z}^\top - \|\boldsymbol{\Sigma}\|_F^2 \cdot \mathbf{I} \right\|_2 \leq \epsilon'_\lambda := C\sigma_u^2\big(n \cdot \|\boldsymbol{\Sigma}\|_2^2 + \sqrt{n} \cdot \|\boldsymbol{\Sigma}^2\|_F\big),$$

where $C$ is an absolute constant.

Based on Lemma A.7, we can give the proof of Lemma 4.4 as follows.

*Proof of Lemma 4.4.* We first derive the lower bound for $I_1$. By Lemma A.5 and the model definition $\mathbf{X} = \mathbf{y}\boldsymbol{\mu}^\top + \mathbf{Q}$, we have

$$\begin{aligned}
\mathbf{y}^\top(\mathbf{X}\mathbf{X}^\top)^{-1}\mathbf{X}\boldsymbol{\mu} &= D^{-1}[(1+\mathbf{y}^\top\mathbf{A}^{-1}\boldsymbol{\nu})\mathbf{y}^\top\mathbf{A}^{-1} - \mathbf{y}^\top\mathbf{A}^{-1}\mathbf{y}\cdot\boldsymbol{\nu}^\top\mathbf{A}^{-1}](\mathbf{y}\boldsymbol{\mu}^\top + \mathbf{Q})\boldsymbol{\mu} \\
&= D^{-1}[(1+\mathbf{y}^\top\mathbf{A}^{-1}\boldsymbol{\nu})\mathbf{y}^\top\mathbf{A}^{-1} - \mathbf{y}^\top\mathbf{A}^{-1}\mathbf{y}\cdot\boldsymbol{\nu}^\top\mathbf{A}^{-1}](\mathbf{y}\cdot\|\boldsymbol{\mu}\|_2^2 + \mathbf{Q}\boldsymbol{\mu}) \\
&= D^{-1}[(1+\mathbf{y}^\top\mathbf{A}^{-1}\boldsymbol{\nu})\mathbf{y}^\top\mathbf{A}^{-1}\mathbf{y} - \mathbf{y}^\top\mathbf{A}^{-1}\mathbf{y}\cdot\boldsymbol{\nu}^\top\mathbf{A}^{-1}\mathbf{y}]\cdot\|\boldsymbol{\mu}\|_2^2 \\
&\quad + D^{-1}[(1+\mathbf{y}^\top\mathbf{A}^{-1}\boldsymbol{\nu})\mathbf{y}^\top\mathbf{A}^{-1}\boldsymbol{\nu} - \mathbf{y}^\top\mathbf{A}^{-1}\mathbf{y}\cdot\boldsymbol{\nu}^\top\mathbf{A}^{-1}\boldsymbol{\nu}]) \\
&= D^{-1}\cdot[(\|\boldsymbol{\mu}\|_2^2 - \boldsymbol{\nu}^\top\mathbf{A}^{-1}\boldsymbol{\nu})\mathbf{y}^\top\mathbf{A}^{-1}\mathbf{y} + (1+\mathbf{y}^\top\mathbf{A}^{-1}\boldsymbol{\nu})\mathbf{y}^\top\mathbf{A}^{-1}\boldsymbol{\nu}], \quad \text{(A.1)}
\end{aligned}$$

where the third equality follows by the notation $\boldsymbol{\nu} = \mathbf{Q}\boldsymbol{\mu}$. By Lemma A.6 and the assumption that $\mathrm{tr}(\boldsymbol{\Sigma}) \geq C\max\{\epsilon_\lambda, n\|\boldsymbol{\Sigma}\|_2, n\|\boldsymbol{\mu}\|_{\boldsymbol{\Sigma}}\}$ for some large enough constant $C$, when $n$ is large enough we have

$$|\mathbf{y}^\top\mathbf{A}^{-1}\boldsymbol{\nu}| \leq \frac{c_1 n}{\mathrm{tr}(\boldsymbol{\Sigma}) - \epsilon_\lambda}\|\boldsymbol{\mu}\|_{\boldsymbol{\Sigma}} \leq \frac{2c_1 n}{\mathrm{tr}(\boldsymbol{\Sigma})}\|\boldsymbol{\mu}\|_{\boldsymbol{\Sigma}} \leq 1,$$

$$0 \leq \boldsymbol{\nu}^\top\mathbf{A}^{-1}\boldsymbol{\nu} \leq \frac{n + c_2\sqrt{n\log(n)}}{\mathrm{tr}(\boldsymbol{\Sigma}) - \epsilon_\lambda}\cdot\|\boldsymbol{\mu}\|_{\boldsymbol{\Sigma}}^2 \leq \frac{2n}{\mathrm{tr}(\boldsymbol{\Sigma})}\cdot\|\boldsymbol{\mu}\|_{\boldsymbol{\Sigma}}^2 \leq \frac{2n\|\boldsymbol{\Sigma}\|_2}{\mathrm{tr}(\boldsymbol{\Sigma})}\cdot\|\boldsymbol{\mu}\|_2^2 \leq \frac{1}{2}\cdot\|\boldsymbol{\mu}\|_2^2,$$

$$\mathbf{y}^\top\mathbf{A}^{-1}\mathbf{y} \geq \frac{n}{\mathrm{tr}(\boldsymbol{\Sigma}) + \epsilon_\lambda} \geq \frac{n}{2\,\mathrm{tr}(\boldsymbol{\Sigma})},$$

where $c_1, c_2$ are absolute constants. Plugging the bounds above into (A.1), we obtain

$$\begin{aligned}
|\mathbf{y}^\top(\mathbf{X}\mathbf{X}^\top)^{-1}\mathbf{X}\boldsymbol{\mu}| &\geq D^{-1}\cdot\left(\frac{1}{2}\cdot\|\boldsymbol{\mu}\|_2^2\cdot\mathbf{y}^\top\mathbf{A}^{-1}\mathbf{y} - 2\cdot|\mathbf{y}^\top\mathbf{A}^{-1}\boldsymbol{\nu}|\right) \\
&\geq D^{-1}\cdot\left[\frac{n}{4\,\mathrm{tr}(\boldsymbol{\Sigma})}\cdot\|\boldsymbol{\mu}\|_2^2 - \frac{4n}{\mathrm{tr}(\boldsymbol{\Sigma})}\|\boldsymbol{\mu}\|_{\boldsymbol{\Sigma}}\right] \\
&\geq D^{-1}\cdot\frac{n}{4\,\mathrm{tr}(\boldsymbol{\Sigma})}\cdot(\|\boldsymbol{\mu}\|_2^2 - 16\|\boldsymbol{\mu}\|_{\boldsymbol{\Sigma}}) \\
&\geq D^{-1}\cdot\frac{n}{8\,\mathrm{tr}(\boldsymbol{\Sigma})}\cdot\|\boldsymbol{\mu}\|_2^2,
\end{aligned}$$

where the last inequality follows by the assumption that $\|\boldsymbol{\mu}\|_2^2 \geq C\|\boldsymbol{\mu}\|_{\boldsymbol{\Sigma}}$ for some large enough absolute constant $C$. Therefore we have

$$[\mathbf{y}^\top(\mathbf{X}\mathbf{X}^\top)^{-1}\mathbf{X}\boldsymbol{\mu}]^2 \geq D^{-2}\cdot\frac{n^2}{64[\mathrm{tr}(\boldsymbol{\Sigma})]^2}\cdot\|\boldsymbol{\mu}\|_2^4 = \frac{H(\boldsymbol{\mu}, \mathbf{Q}, \mathbf{y}, \boldsymbol{\Sigma})}{64}\cdot n^2\|\boldsymbol{\mu}\|_2^4,$$

where we define

$$H(\boldsymbol{\mu}, \mathbf{Q}, \mathbf{y}, \boldsymbol{\Sigma}) := [D\cdot\mathrm{tr}(\boldsymbol{\Sigma})]^{-2} > 0.$$

This completes the proof of the lower bound of $I_1$.

For $I_2$, by Lemma A.5 we have

$$
\begin{aligned}
\mathbf{y}^\top (\mathbf{X}\mathbf{X}^\top)^{-1}\mathbf{y} &= D^{-1}[(1 + \mathbf{y}^\top \mathbf{A}^{-1}\boldsymbol{\nu})\mathbf{y}^\top \mathbf{A}^{-1}\mathbf{y} - \mathbf{y}^\top \mathbf{A}^{-1}\mathbf{y} \cdot \boldsymbol{\nu}^\top \mathbf{A}^{-1}\mathbf{y}] \\
&= D^{-1}[(1 + \mathbf{y}^\top \mathbf{A}^{-1}\boldsymbol{\nu})\mathbf{y}^\top \mathbf{A}^{-1}\mathbf{y} - \mathbf{y}^\top \mathbf{A}^{-1}\mathbf{y} \cdot \boldsymbol{\nu}^\top \mathbf{A}^{-1}\mathbf{y}] \\
&= D^{-1} \cdot \mathbf{y}^\top \mathbf{A}^{-1}\mathbf{y} \\
&\leq D^{-1} \cdot \frac{n}{\mathrm{tr}(\boldsymbol{\Sigma}) - \epsilon_\lambda} \\
&\leq 2D^{-1} \cdot \frac{n}{\mathrm{tr}(\boldsymbol{\Sigma})},
\end{aligned}
$$

where the first inequality follows by Lemma A.6, and the second inequality follows by the assumption that $\mathrm{tr}(\boldsymbol{\Sigma}) \geq C\epsilon_\lambda$ for some large enough constant $C$. Therefore we have

$$
I_2 = (\mathbf{y}^\top (\mathbf{X}\mathbf{X}^\top)^{-1}\mathbf{y})^2 \cdot \|\boldsymbol{\mu}\|_{\boldsymbol{\Sigma}}^2 \leq 4D^{-2} \cdot \frac{n^2 \cdot \|\boldsymbol{\mu}\|_{\boldsymbol{\Sigma}}^2}{[\mathrm{tr}(\boldsymbol{\Sigma})]^2} = 4H(\boldsymbol{\mu}, \mathbf{Q}, \mathbf{y}, \boldsymbol{\Sigma}) \cdot n^2 \cdot \|\boldsymbol{\mu}\|_{\boldsymbol{\Sigma}}^2,
$$

where we use the definition $H(\boldsymbol{\mu}, \mathbf{Q}, \mathbf{y}, \boldsymbol{\Sigma}) = [D \cdot \mathrm{tr}(\boldsymbol{\Sigma})]^{-2}$. This proves the upper bound of $I_2$.

For $I_3$, by our calculation in Lemma A.5, we have

$$
\mathbf{y}^\top (\mathbf{X}\mathbf{X}^\top)^{-1} = D^{-1}[(1 + \mathbf{y}^\top \mathbf{A}^{-1}\boldsymbol{\nu})\mathbf{y} - \mathbf{y}^\top \mathbf{A}^{-1}\mathbf{y} \cdot \boldsymbol{\nu}]^\top \mathbf{A}^{-1}.
$$

Denote $\mathbf{a} = D^{-1}[(1 + \mathbf{y}^\top \mathbf{A}^{-1}\boldsymbol{\nu}) \cdot \mathbf{y} - \mathbf{y}^\top \mathbf{A}^{-1}\mathbf{y} \cdot \boldsymbol{\nu}]$. Then

$$
\begin{aligned}
I_3 &= \mathbf{y}^\top (\mathbf{X}\mathbf{X}^\top)^{-1}\mathbf{Q}\boldsymbol{\Sigma}\mathbf{Q}^\top (\mathbf{X}\mathbf{X}^\top)^{-1}\mathbf{y} \\
&= \mathbf{a}^\top (\mathbf{Q}\mathbf{Q}^\top)^{-1}\mathbf{Q}\boldsymbol{\Sigma}\mathbf{Q}^\top (\mathbf{Q}\mathbf{Q}^\top)^{-1}\mathbf{a} \\
&= \mathbf{a}^\top (\mathbf{Z}\boldsymbol{\Lambda}\mathbf{Z}^\top)^{-1}\mathbf{Z}\boldsymbol{\Lambda}^2\mathbf{Z}^\top (\mathbf{Z}\boldsymbol{\Lambda}\mathbf{Z}^\top)^{-1}\mathbf{a},
\end{aligned} \tag{A.2}
$$

where we plug in $\boldsymbol{\Sigma} = \mathbf{V}\boldsymbol{\Lambda}\mathbf{V}^\top$ and $\mathbf{Q} = \mathbf{Z}\boldsymbol{\Lambda}^{1/2}\mathbf{V}^\top$ for $\mathbf{Z}$ with independent sub-Gaussian entries. By Lemma A.4, Lemma A.7 and (A.2), when $\mathrm{tr}(\boldsymbol{\Sigma}) \geq \epsilon_\lambda$ we have

$$
\begin{aligned}
I_3 &= \mathbf{a}^\top (\mathbf{Z}\boldsymbol{\Lambda}\mathbf{Z}^\top)^{-1}\mathbf{Z}\boldsymbol{\Lambda}^2\mathbf{Z}^\top (\mathbf{Z}\boldsymbol{\Lambda}\mathbf{Z}^\top)^{-1}\mathbf{a} \\
&\leq \mathbf{a}^\top (\mathbf{Z}\boldsymbol{\Lambda}\mathbf{Z}^\top)^{-2}\mathbf{a} \cdot \left[ \|\boldsymbol{\Sigma}\|_F^2 + \epsilon_\lambda' \right] \\
&\leq \|\mathbf{a}\|_2^2 \cdot \frac{\|\boldsymbol{\Sigma}\|_F^2 + \epsilon_\lambda'}{[\mathrm{tr}(\boldsymbol{\Sigma}) - \epsilon_\lambda]^2}.
\end{aligned} \tag{A.3}
$$

Here the first inequality follows by Lemma A.7, and the second inequality follows by Lemma A.4. By definition, we have

$$
\begin{aligned}
\|\mathbf{a}\|_2^2 &= \|D^{-1}(1 + \mathbf{y}^\top \mathbf{A}^{-1}\boldsymbol{\nu})\mathbf{y} - \mathbf{y}^\top \mathbf{A}^{-1}\mathbf{y} \cdot \boldsymbol{\nu}\|_2^2 \\
&\leq 2D^{-2}(1 + \mathbf{y}^\top \mathbf{A}^{-1}\boldsymbol{\nu})^2\|\mathbf{y}\|_2^2 + 2D^{-2}(\mathbf{y}^\top \mathbf{A}^{-1}\mathbf{y})^2 \cdot \|\mathbf{Q}\boldsymbol{\mu}\|_2^2.
\end{aligned}
$$

Then with the same proof as in Lemma A.6, when $n$ is sufficiently large, with probability at least $1 - O(n^{-2})$ we have

$$
\|\mathbf{Q}\boldsymbol{\mu}\|_2^2 \leq 2n\|\boldsymbol{\mu}\|_{\boldsymbol{\Sigma}}^2.
$$

Therefore we have

$$
\begin{aligned}
\|\mathbf{a}\|_2^2 &\leq 2D^{-2}(1 + \mathbf{y}^\top \mathbf{A}^{-1}\boldsymbol{\nu})^2\|\mathbf{y}\|_2^2 + 2D^{-2}(\mathbf{y}^\top \mathbf{A}^{-1}\mathbf{y})^2 \cdot \|\mathbf{Q}\boldsymbol{\mu}\|_2^2 \\
&\leq 2D^{-2}(1 + \mathbf{y}^\top \mathbf{A}^{-1}\boldsymbol{\nu})^2 \cdot n + 4D^{-2}(\mathbf{y}^\top \mathbf{A}^{-1}\mathbf{y})^2 \cdot n \cdot \|\boldsymbol{\mu}\|_{\boldsymbol{\Sigma}}^2.
\end{aligned} \tag{A.4}
$$

Moreover, by Lemma A.6 and the assumption that $\mathrm{tr}(\boldsymbol{\Sigma}) \geq C \max\{\epsilon_\lambda, n, n\|\boldsymbol{\mu}\|_{\boldsymbol{\Sigma}}\}$ for some large enough constant $C$, we have

$$
\begin{aligned}
|\mathbf{y}^\top \mathbf{A}^{-1}\boldsymbol{\nu}| &\leq \frac{c_3 n}{\mathrm{tr}(\boldsymbol{\Sigma}) - \epsilon_\lambda}\|\boldsymbol{\mu}\|_{\boldsymbol{\Sigma}} \leq \sqrt{2} - 1, \\
\mathbf{y}^\top \mathbf{A}^{-1}\mathbf{y} &\leq \frac{n}{\mathrm{tr}(\boldsymbol{\Sigma}) - \epsilon_\lambda} \leq \frac{2n}{\mathrm{tr}(\boldsymbol{\Sigma})},
\end{aligned}
$$

where $c_3$ is an absolute constant. Plugging the above bounds into (A.4), we obtain

$$\|\mathbf{a}\|_2^2 \le 2D^{-2}(1 + \mathbf{y}^\top \mathbf{A}^{-1}\boldsymbol{\nu})^2 \cdot n + 4D^{-2}(\mathbf{y}^\top \mathbf{A}^{-1}\mathbf{y})^2 \cdot n \cdot \|\boldsymbol{\mu}\|_{\boldsymbol{\Sigma}}^2$$

$$\le 4D^{-2} \cdot n + 8D^{-2} \cdot n \cdot \left[\frac{n}{\text{tr}(\boldsymbol{\Sigma})} \cdot \|\boldsymbol{\mu}\|_{\boldsymbol{\Sigma}}\right]^2$$

$$\le 5D^{-2} \cdot n,$$

where the last inequality utilizes the assumption $\text{tr}(\boldsymbol{\Sigma}) \ge Cn\|\boldsymbol{\mu}\|_{\boldsymbol{\Sigma}}$ for some large enough constant $C$ again. Further plugging this bound into (A.3), we obtain

$$I_3 \le \|\mathbf{a}\|_2^2 \cdot \frac{\|\boldsymbol{\Sigma}\|_F^2 + \epsilon_\lambda'}{[\text{tr}(\boldsymbol{\Sigma}) - \epsilon_\lambda]^2} \le 5D^{-2}n \cdot \frac{\|\boldsymbol{\Sigma}\|_F^2 + \epsilon_\lambda'}{[\text{tr}(\boldsymbol{\Sigma}) - \epsilon_\lambda]^2}$$

$$\le c_4 D^{-2} \cdot \frac{n \cdot \|\boldsymbol{\Sigma}\|_F^2 + n^2 \cdot \|\boldsymbol{\Sigma}\|_2^2 + n^{3/2} \cdot \|\boldsymbol{\Sigma}^2\|_F}{[\text{tr}(\boldsymbol{\Sigma})]^2}, \tag{A.5}$$

where $c_4$ is an absolute constant. Note that we have

$$n^{3/2} \cdot \|\boldsymbol{\Sigma}^2\|_F \le n \cdot \|\boldsymbol{\Sigma}\|_F \cdot (\sqrt{n} \cdot \|\boldsymbol{\Sigma}\|_2) \le n \cdot (\|\boldsymbol{\Sigma}\|_F^2 + n \cdot \|\boldsymbol{\Sigma}\|_2^2)/2.$$

Plugging this bound into (A.5), we have

$$I_3 \le c_5 D^{-2} \cdot \frac{n \cdot \|\boldsymbol{\Sigma}\|_F^2 + n^2 \cdot \|\boldsymbol{\Sigma}\|_2^2}{[\text{tr}(\boldsymbol{\Sigma})]^2} = c_5 H(\boldsymbol{\mu}, \mathbf{Q}, \mathbf{y}, \boldsymbol{\Sigma}) \cdot (n \cdot \|\boldsymbol{\Sigma}\|_F^2 + n^2 \cdot \|\boldsymbol{\Sigma}\|_2^2),$$

where we use the definition $H(\boldsymbol{\mu}, \mathbf{Q}, \mathbf{y}, \boldsymbol{\Sigma}) = [D \cdot \text{tr}(\boldsymbol{\Sigma})]^{-2}$, and $c_4$ is an absolute constant. This finishes the proof of the upper bound of $I_3$. $\square$

### A.3  Proof of Lemmas in Appendix A.1

We present the proofs of Lemmas A.2 and A.3.

#### A.3.1  Proof of Lemma A.2

Here we present the proof of Lemma A.2. The proof utilizes Lemma A.5 and an argument based on the polarization identity.

*Proof of Lemma A.2.* By Lemma A.5, we have

$$\mathbf{y}^\top (\mathbf{X}\mathbf{X}^\top)^{-1}\mathbf{e}_i y_i = D^{-1}[(1 + \mathbf{y}^\top \mathbf{A}^{-1}\boldsymbol{\nu})\mathbf{y}^\top \mathbf{A}^{-1}\mathbf{e}_i y_i - \mathbf{y}^\top \mathbf{A}^{-1}\mathbf{y} \cdot \boldsymbol{\nu}^\top \mathbf{A}^{-1}\mathbf{e}_i y_i]. \tag{A.6}$$

Moreover, by definition we have

$$\mathbf{y}^\top \mathbf{A}^{-1}\mathbf{e}_i y_i = \frac{1}{4\sqrt{n}}(\mathbf{y} + \sqrt{n}\mathbf{e}_i y_i)^\top \mathbf{A}^{-1}(\mathbf{y} + \sqrt{n}\mathbf{e}_i y_i) - \frac{1}{4\sqrt{n}}(\mathbf{y} - \sqrt{n}\mathbf{e}_i y_i)^\top \mathbf{A}^{-1}(\mathbf{y} - \sqrt{n}\mathbf{e}_i y_i)$$

$$\ge \frac{1}{4\sqrt{n}}\left[\frac{\|\mathbf{y} + \sqrt{n}\mathbf{e}_i y_i\|_2^2}{\text{tr}(\boldsymbol{\Sigma}) + \epsilon_\lambda} - \frac{\|\mathbf{y} - \sqrt{n}\mathbf{e}_i y_i\|_2^2}{\text{tr}(\boldsymbol{\Sigma}) - \epsilon_\lambda}\right]$$

$$= \frac{1}{4\sqrt{n}}\left[\frac{2n + 2\sqrt{n}}{\text{tr}(\boldsymbol{\Sigma}) + \epsilon_\lambda} - \frac{2n - 2\sqrt{n}}{\text{tr}(\boldsymbol{\Sigma}) - \epsilon_\lambda}\right]$$

$$= \frac{1}{2\sqrt{n}} \cdot \frac{(n + \sqrt{n})(\text{tr}(\boldsymbol{\Sigma}) - \epsilon_\lambda) - (n - \sqrt{n})(\text{tr}(\boldsymbol{\Sigma}) + \epsilon_\lambda)}{\text{tr}(\boldsymbol{\Sigma})^2 - \epsilon_\lambda^2}$$

$$= \frac{1}{2\sqrt{n}} \cdot \frac{2\sqrt{n}\,\text{tr}(\boldsymbol{\Sigma}) - 2n\epsilon_\lambda}{\text{tr}(\boldsymbol{\Sigma})^2 - \epsilon_\lambda^2}$$

$$= \frac{\text{tr}(\boldsymbol{\Sigma}) - \sqrt{n}\epsilon_\lambda}{\text{tr}(\boldsymbol{\Sigma})^2 - \epsilon_\lambda^2}, \tag{A.7}$$

where we use the polarization identity $\mathbf{a}^\top \mathbf{M}\mathbf{b} = 1/4(\mathbf{a} + \mathbf{b})^\top \mathbf{M}(\mathbf{a} + \mathbf{b}) - 1/4(\mathbf{a} - \mathbf{b})^\top \mathbf{M}(\mathbf{a} - \mathbf{b})$ in the first equality and use Lemma A.4 to derive the inequality.

Plugging (A.7) and the inequalities in Lemmas A.6 into (A.6), we have that as long as $\mathrm{tr}(\boldsymbol{\Sigma}) > c_1 \max\{n\|\boldsymbol{\mu}\|_{\boldsymbol{\Sigma}}, \epsilon_\lambda\}$ for some large enough constant $c_1$, $\mathbf{y}^\top \mathbf{A}^{-1}\boldsymbol{\nu} \leq 1/2$ and therefore

$$
\begin{aligned}
\mathbf{y}^\top (\mathbf{X}\mathbf{X}^\top)^{-1}\mathbf{e}_i y_i &= D^{-1}[(1 + \mathbf{y}^\top \mathbf{A}^{-1}\boldsymbol{\nu})\mathbf{y}^\top \mathbf{A}^{-1}\mathbf{e}_i y_i - \mathbf{y}^\top \mathbf{A}^{-1}\mathbf{y} \cdot \boldsymbol{\nu}^\top \mathbf{A}^{-1}\mathbf{e}_i y_i] \\
&\geq D^{-1} \cdot \left[ \frac{1}{2} \cdot \mathbf{y}^\top \mathbf{A}^{-1}\mathbf{e}_i y_i - \frac{c_2 n}{\mathrm{tr}(\boldsymbol{\Sigma})} \cdot |\boldsymbol{\nu}^\top \mathbf{A}^{-1}\mathbf{e}_i y_i| \right],
\end{aligned} \tag{A.8}
$$

where $c_2$ is an absolute constant. By (A.7), we can see that as long as $\mathrm{tr}(\boldsymbol{\Sigma}) \geq c_3 \sqrt{n}\epsilon_\lambda$ for some large enough absolute constant $c_3$, we have

$$
\mathbf{y}^\top \mathbf{A}^{-1}\mathbf{e}_i y_i \geq \frac{\mathrm{tr}(\boldsymbol{\Sigma}) - \sqrt{n}\epsilon_\lambda}{\mathrm{tr}(\boldsymbol{\Sigma})^2 - \epsilon_\lambda^2} \geq \frac{1}{2\,\mathrm{tr}(\boldsymbol{\Sigma})}.
$$

Plugging the bound above into (A.8), we obtain

$$
\mathbf{y}^\top (\mathbf{X}\mathbf{X}^\top)^{-1}\mathbf{e}_i y_i \geq \frac{1}{4D\,\mathrm{tr}(\boldsymbol{\Sigma})} \cdot [1 - c_4 n \cdot |\boldsymbol{\nu}^\top \mathbf{A}^{-1}\mathbf{e}_i y_i|].
$$

Since $D > 0$, we see that $G(\boldsymbol{\mu}, \mathbf{Q}, \mathbf{y}, \boldsymbol{\Sigma}) := [4D\,\mathrm{tr}(\boldsymbol{\Sigma})]^{-1} > 0$. This completes the proof. $\qquad \square$

### A.3.2 Proof of Lemma A.3

Here we give the detailed proof of Lemma A.3 to backup the proof sketch presented in Section 4. The proof is based on the polarization identity.

*Proof of Lemma A.3.* We have the following calculation,

$$
\begin{aligned}
\boldsymbol{\mu}^\top \mathbf{Q}^\top \mathbf{A}^{-1}\mathbf{e}_i y_i &= \frac{1}{\|\mathbf{Q}\boldsymbol{\mu}\|_2} \cdot (\mathbf{Q}\boldsymbol{\mu})^\top \mathbf{A}^{-1}(\|\mathbf{Q}\boldsymbol{\mu}\|_2 \cdot \mathbf{e}_i y_i) \\
&= \frac{1}{4\|\mathbf{Q}\boldsymbol{\mu}\|_2} \cdot (\mathbf{Q}\boldsymbol{\mu} + \|\mathbf{Q}\boldsymbol{\mu}\|_2 \cdot \mathbf{e}_i y_i)^\top \mathbf{A}^{-1}(\mathbf{Q}\boldsymbol{\mu} + \|\mathbf{Q}\boldsymbol{\mu}\|_2 \cdot \mathbf{e}_i y_i) \\
&\quad - \frac{1}{4\|\mathbf{Q}\boldsymbol{\mu}\|_2} \cdot (\mathbf{Q}\boldsymbol{\mu} - \|\mathbf{Q}\boldsymbol{\mu}\|_2 \cdot \mathbf{e}_i y_i)^\top \mathbf{A}^{-1}(\mathbf{Q}\boldsymbol{\mu} - \|\mathbf{Q}\boldsymbol{\mu}\|_2 \cdot \mathbf{e}_i y_i) \\
&\leq \frac{1}{4\|\mathbf{Q}\boldsymbol{\mu}\|_2} \cdot \left[ \frac{\|\mathbf{Q}\boldsymbol{\mu} + \|\mathbf{Q}\boldsymbol{\mu}\|_2 \cdot \mathbf{e}_i y_i\|_2^2}{\mathrm{tr}(\boldsymbol{\Sigma}) - \epsilon_\lambda} - \frac{\|\mathbf{Q}\boldsymbol{\mu} - \|\mathbf{Q}\boldsymbol{\mu}\|_2 \cdot \mathbf{e}_i y_i\|_2^2}{\mathrm{tr}(\boldsymbol{\Sigma}) + \epsilon_\lambda} \right] \\
&= \frac{1}{4\|\mathbf{Q}\boldsymbol{\mu}\|_2} \cdot \left[ \frac{2\|\mathbf{Q}\boldsymbol{\mu}\|_2^2 + 2y_i\|\mathbf{Q}\boldsymbol{\mu}\|_2 \cdot \mathbf{e}_i^\top \mathbf{Q}\boldsymbol{\mu}}{\mathrm{tr}(\boldsymbol{\Sigma}) - \epsilon_\lambda} - \frac{2\|\mathbf{Q}\boldsymbol{\mu}\|_2^2 - 2y_i\|\mathbf{Q}\boldsymbol{\mu}\|_2 \cdot \mathbf{e}_i^\top \mathbf{Q}\boldsymbol{\mu}}{\mathrm{tr}(\boldsymbol{\Sigma}) + \epsilon_\lambda} \right] \\
&= \frac{1}{2\|\mathbf{Q}\boldsymbol{\mu}\|_2} \cdot \frac{2\|\mathbf{Q}\boldsymbol{\mu}\|_2^2 \cdot \epsilon_\lambda + 2y_i\|\mathbf{Q}\boldsymbol{\mu}\|_2 \cdot \mathbf{e}_i^\top \mathbf{Q}\boldsymbol{\mu} \cdot \mathrm{tr}(\boldsymbol{\Sigma})}{\mathrm{tr}(\boldsymbol{\Sigma})^2 - \epsilon_\lambda^2} \\
&= \frac{\|\mathbf{Q}\boldsymbol{\mu}\|_2 \cdot \epsilon_\lambda + y_i\mathbf{e}_i^\top \mathbf{Q}\boldsymbol{\mu} \cdot \mathrm{tr}(\boldsymbol{\Sigma})}{\mathrm{tr}(\boldsymbol{\Sigma})^2 - \epsilon_\lambda^2},
\end{aligned} \tag{A.9}
$$

where the first equality holds due to the polarization identity $\mathbf{a}^\top \mathbf{M}\mathbf{b} = 1/4(\mathbf{a} + \mathbf{b})^\top \mathbf{M}(\mathbf{a} + \mathbf{b}) - 1/4(\mathbf{a} - \mathbf{b})^\top \mathbf{M}(\mathbf{a} - \mathbf{b})$, and the first inequality follows by Lemma A.4. Based on our model assumption, we can denote $\mathbf{Q} = \mathbf{Z}\boldsymbol{\Lambda}^{1/2}\mathbf{V}^\top$, where the entries of $\mathbf{Z}$ are independent sub-Gaussian random variables with $\|\mathbf{Z}_{ij}\|_{\psi_2} \leq \sigma_u$ for all $i \in [n]$ and $j \in [p]$. Denote $\widetilde{\boldsymbol{\mu}} = \boldsymbol{\Lambda}^{1/2}\mathbf{V}^\top \boldsymbol{\mu}$. Then with the same proof as in Lemma A.6, we have

$$
\|\mathbf{Q}\boldsymbol{\mu}\|_2^2 = \|\mathbf{Z}\widetilde{\boldsymbol{\mu}}\|_2^2 \leq 2n\|\widetilde{\boldsymbol{\mu}}\|_2^2 = 2n\|\boldsymbol{\mu}\|_{\boldsymbol{\Sigma}}^2
$$

when $n$ is large enough. Moreover, we also have

$$
\|y_i\mathbf{e}_i^\top \mathbf{Q}\boldsymbol{\mu}\|_{\psi_2} = \left\| \sum_{j=1}^p \mathbf{Z}_{ij}\widetilde{\mu}_j \right\|_{\psi_2} \leq \|\widetilde{\boldsymbol{\mu}}\|_2 \cdot \sigma_u.
$$

Therefore by Hoeffding's inequality, with probability at least $1 - n^{-1}$, we have

$$
|y_i\mathbf{e}_i^\top \mathbf{Q}\boldsymbol{\mu}| \leq c_1\|\widetilde{\boldsymbol{\mu}}\|_2 \cdot \sqrt{\log(n)} = c_1\|\boldsymbol{\mu}\|_{\boldsymbol{\Sigma}} \cdot \sqrt{\log(n)},
$$

where $c_1$ is an absolute constant. Therefore we have

$$\boldsymbol{\nu}^\top \mathbf{A}^{-1} \mathbf{e}_i y_i \leq \frac{\sqrt{2n}\|\boldsymbol{\mu}\|_{\boldsymbol{\Sigma}} \cdot \epsilon_\lambda + c_2 \|\boldsymbol{\mu}\|_{\boldsymbol{\Sigma}} \sqrt{\log(n)} \cdot \mathrm{tr}(\boldsymbol{\Sigma})}{\mathrm{tr}(\boldsymbol{\Sigma})^2 - \epsilon_\lambda^2}.$$

With the exact same proof, we also have

$$-\boldsymbol{\nu}^\top \mathbf{A}^{-1} \mathbf{e}_i y_i \leq \frac{\sqrt{2n}\|\boldsymbol{\mu}\|_{\boldsymbol{\Sigma}} \cdot \epsilon_\lambda + c_2 \|\boldsymbol{\mu}\|_{\boldsymbol{\Sigma}} \sqrt{\log(n)} \cdot \mathrm{tr}(\boldsymbol{\Sigma})}{\mathrm{tr}(\boldsymbol{\Sigma})^2 - \epsilon_\lambda^2}.$$

Therefore by the assumption that $\mathrm{tr}(\boldsymbol{\Sigma}) > C\sqrt{n}\epsilon_\lambda$ for some large enough absolute constant $C$, we have

$$\left|\boldsymbol{\nu}^\top \mathbf{A}^{-1} \mathbf{e}_i\right| \leq \frac{c_3 \|\boldsymbol{\mu}\|_{\boldsymbol{\Sigma}} \cdot \sqrt{\log(n)}}{\mathrm{tr}(\boldsymbol{\Sigma})}$$

for some absolute constant $c_3$. This completes the proof. $\qquad \square$

## A.4 Proof of Lemmas in Appendix A.2

Here we present the proofs of lemmas we used in Appendix A.2.

### A.4.1 Proof of Lemma A.4

The proof of Lemma A.4 is motivated by the analysis given in [2]. However here in Lemma A.4 we give a slightly tighter bound. The proof is as follows.

*Proof of Lemma A.4.* Let $\mathcal{N}$ be a $1/4$-net on the unit sphere $s^{n-1}$. Then by Lemma 5.2 in [24], we have $|\mathcal{N}| \leq 9^n$. Denote $\mathbf{z}_j = \lambda_j^{-1/2} \mathbf{Q} \mathbf{v}_j \in \mathbb{R}^n$. Then by definition, for any fixed unit vector $\widehat{\mathbf{a}} \in \mathcal{N}$ we have $\widehat{\mathbf{a}}^\top \mathbf{A} \widehat{\mathbf{a}} = \mathbf{Q}\mathbf{Q}^\top = \widehat{\mathbf{a}}^\top \sum_{j=1}^p \lambda_j \mathbf{z}_j \mathbf{z}_j^\top \widehat{\mathbf{a}} = \sum_{j=1}^p \lambda_j (\widehat{\mathbf{a}}^\top \mathbf{z}_j)^2$. By Lemma 5.9 in [24], there exists an absolute constant $c_1$ such that $\|\widehat{\mathbf{a}}^\top \mathbf{z}_j\|_{\psi_2} \leq c_1 \sigma_u$. Therefore by Lemma 21 and Corollary 23 in [2], for any $t > 0$, with probability at least $1 - 2\exp(-t)$ we have

$$\left|\widehat{\mathbf{a}}^\top \mathbf{A} \widehat{\mathbf{a}} - \mathrm{tr}(\boldsymbol{\Sigma})\right| \leq c_2 \sigma_u^2 \max\left(t \cdot \|\boldsymbol{\Sigma}\|_2, \sqrt{t} \cdot \|\boldsymbol{\Sigma}\|_F\right).$$

Applying an union bound over all $\widehat{\mathbf{a}} \in \mathcal{N}$, we have that with probability at least $1 - 2 \cdot 9^n \exp(-t)$,

$$\left|\widehat{\mathbf{a}}^\top \mathbf{A} \widehat{\mathbf{a}} - \mathrm{tr}(\boldsymbol{\Sigma})\right| \leq c_2 \sigma_u^2 \max\left(t \cdot \|\boldsymbol{\Sigma}\|_2, \sqrt{t} \cdot \|\boldsymbol{\Sigma}\|_F\right)$$

for all $\widehat{\mathbf{a}} \in \mathcal{N}$. Therefore by Lemma 25 in [2], with probability at least $1 - 2 \cdot 9^n \exp(-t)$, we have

$$\left\|\mathbf{A} - \mathrm{tr}(\boldsymbol{\Sigma})\mathbf{I}\right\|_2 \leq c_3 \sigma_u^2 \left(t \cdot \|\boldsymbol{\Sigma}\|_2 + \sqrt{t} \cdot \|\boldsymbol{\Sigma}\|_F\right),$$

where $c_3$ is an absolute constant. Setting $t = c_4 n$ for some large enough constant $c_4$, we have that with probability at least $1 - n^{-2}$,

$$\left\|\mathbf{A} - \mathrm{tr}(\boldsymbol{\Sigma})\mathbf{I}\right\|_2 \leq c_5 \sigma_u^2 \left(n \cdot \|\boldsymbol{\Sigma}\|_2 + \sqrt{n} \cdot \|\boldsymbol{\Sigma}\|_F\right),$$

where $c_5$ is an absolute constant. This completes the proof. $\qquad \square$

### A.4.2 Proof of Lemma A.5

Here we present the proof of Lemma A.5. Our proof utilizes a key lemma by [25], and gives further simplifications of the result.

*Proof of Lemma A.5.* Denote $s = \mathbf{y}^\top \mathbf{A}^{-1} \mathbf{y}$, $t = \boldsymbol{\nu}^\top \mathbf{A}^{-1} \boldsymbol{\nu}$, $h = \mathbf{y}^\top \mathbf{A}^{-1} \boldsymbol{\nu}$. Then we have $D = \|\boldsymbol{\mu}\|_2^2 s - st + (h+1)^2$. By Lemma 3 in [25], we have

$$\mathbf{y}^\top (\mathbf{X}\mathbf{X}^\top)^{-1} = \mathbf{y}^\top \mathbf{A}^{-1} - D^{-1} \cdot [\|\boldsymbol{\mu}\|_2^2 s + h^2 + h - st] \cdot \mathbf{y}^\top \mathbf{A}^{-1} - D^{-1} s \cdot \boldsymbol{\nu}^\top \mathbf{A}^{-1}.$$

Rearranging terms, we obtain

$$\mathbf{y}^\top(\mathbf{X}\mathbf{X}^\top)^{-1} = \left[1 - \frac{\|\boldsymbol{\mu}\|_2^2 s + h^2 + h - st}{\|\boldsymbol{\mu}\|_2^2 s - st + (h+1)^2}\right] \cdot \mathbf{y}^\top\mathbf{A}^{-1} - D^{-1}s \cdot \boldsymbol{\nu}^\top\mathbf{A}^{-1}$$

$$= \frac{h+1}{\|\boldsymbol{\mu}\|_2^2 s - st + (h+1)^2} \cdot \mathbf{y}^\top\mathbf{A}^{-1} - D^{-1}s \cdot \boldsymbol{\nu}^\top\mathbf{A}^{-1}$$

$$= D^{-1}[(h+1)\mathbf{y}^\top\mathbf{A}^{-1} - s \cdot \boldsymbol{\nu}^\top\mathbf{A}^{-1}].$$

At last, by the definition of $D$, we have

$$D = \mathbf{y}^\top\mathbf{A}^{-1}\mathbf{y} \cdot (\|\boldsymbol{\mu}\|_2^2 - \boldsymbol{\mu}^\top\mathbf{Q}^\top(\mathbf{Q}\mathbf{Q}^\top)^{-1}\mathbf{Q}\boldsymbol{\mu}) + (1 + \mathbf{y}^\top\mathbf{A}^{-1}\boldsymbol{\nu})^2$$

$$\geq (1 + \mathbf{y}^\top\mathbf{A}^{-1}\boldsymbol{\nu})^2,$$

where we utilize the fact that $\mathbf{y}^\top\mathbf{A}^{-1}\mathbf{y} \geq 0$ and $\|\boldsymbol{\mu}\|_2^2 \geq \boldsymbol{\mu}^\top\mathbf{Q}^\top(\mathbf{Q}\mathbf{Q}^\top)^{-1}\mathbf{Q}\boldsymbol{\mu}$. Since $\mathbf{y}^\top\mathbf{A}^{-1}\boldsymbol{\nu} \neq 1$ with probability 1, we see that $D > 0$ almost surely. This completes the proof. $\qquad\square$

### A.4.3 Proof of Lemma A.6

The proof of Lemma A.6 is based on the application of eigenvalue concentration results in Lemma A.4. We present the details as follows.

*Proof of Lemma A.6.* The bounds on $\mathbf{y}^\top\mathbf{A}^{-1}\mathbf{y}$ are directly derived from Lemma A.4 and the fact that $\|\mathbf{y}\|_2^2 = n$. To derive the bounds for $\boldsymbol{\nu}\mathbf{A}^{-1}\boldsymbol{\nu}$, we note that by definition, $\boldsymbol{\nu} = \mathbf{Q}\boldsymbol{\mu}$ and

$$\boldsymbol{\nu}^\top\mathbf{A}^{-1}\boldsymbol{\nu} = \boldsymbol{\mu}^\top\mathbf{Q}^\top(\mathbf{Q}\mathbf{Q}^\top)^{-1}\mathbf{Q}\boldsymbol{\mu}.$$

Denote $\mathbf{z}_i = \lambda_i^{-1/2}\mathbf{Q}\mathbf{v}_i \in \mathbb{R}^n$, $\mathbf{Z} = [\mathbf{z}_1, \ldots, \mathbf{z}_p] \in \mathbb{R}^{n \times p}$, and $\widetilde{\boldsymbol{\mu}} = \Lambda^{1/2}\mathbf{V}^\top\boldsymbol{\mu}$. Then $\mathbf{Q} = \mathbf{Z}\Lambda^{1/2}\mathbf{V}^\top$, $\mathbf{Q}\boldsymbol{\mu} = \mathbf{Z}\widetilde{\boldsymbol{\mu}}$, and

$$\boldsymbol{\mu}^\top\mathbf{Q}^\top(\mathbf{Q}\mathbf{Q}^\top)^{-1}\mathbf{Q}\boldsymbol{\mu} = \boldsymbol{\mu}^\top\mathbf{V}\Lambda^{1/2}\mathbf{Z}^\top(\mathbf{Z}\Lambda\mathbf{Z}^\top)^{-1}\mathbf{Z}\Lambda^{1/2}\mathbf{V}^\top\boldsymbol{\mu}$$

$$= \widetilde{\boldsymbol{\mu}}^\top\mathbf{Z}^\top(\mathbf{Z}\Lambda\mathbf{Z}^\top)^{-1}\mathbf{Z}\widetilde{\boldsymbol{\mu}}$$

$$\leq \frac{\|\mathbf{Z}\widetilde{\boldsymbol{\mu}}\|_2^2}{\text{tr}(\boldsymbol{\Sigma}) - \epsilon_\lambda}.$$

Similarly, we have

$$\boldsymbol{\mu}^\top\mathbf{Q}^\top(\mathbf{Q}\mathbf{Q}^\top)^{-1}\mathbf{Q}\boldsymbol{\mu} \geq \frac{\|\mathbf{Z}\widetilde{\boldsymbol{\mu}}\|_2^2}{\text{tr}(\boldsymbol{\Sigma}) + \epsilon_\lambda}.$$

We now proceed to give upper and lower bounds for the term $\|\mathbf{Z}\widetilde{\boldsymbol{\mu}}\|_2^2 = \sum_{i=1}^n (\sum_{j=1}^p \mathbf{Z}_{ij}\widetilde{\mu}_j)^2$. Note that by definition, $\mathbf{Z}_{ij}$ for $i \in [n]$ and $j \in [p]$ are independent sub-Gaussian vectors with $\|\mathbf{Z}_{ij}\|_{\psi_2} \leq \sigma_u$. By Lemma 5.9 in [24], we have

$$\left\|\sum_{j=1}^p \mathbf{Z}_{ij}\widetilde{\mu}_j\right\|_{\psi_2} \leq c_1\|\widetilde{\boldsymbol{\mu}}\|_2 \cdot \sigma_u,$$

where $c_1$ is an absolute constant. Therefore by Lemma 5.14 in [24], we have

$$\left\|\left(\sum_{j=1}^p \mathbf{Z}_{ij}\widetilde{\mu}_j\right)^2 - \|\widetilde{\boldsymbol{\mu}}\|_2^2\right\|_{\psi_1} \leq c_2\|\widetilde{\mu}_j\|_2^2,$$

where we merge $\sigma_u$ into the absolute constant $c_2$. By Bernstein's inequality, with probability at least $1 - n^{-2}$,

$$\left|\|\mathbf{Z}\widetilde{\boldsymbol{\mu}}\|_2^2 - \mathbb{E}\|\mathbf{Z}\widetilde{\boldsymbol{\mu}}\|_2^2\right| \leq c_3\|\widetilde{\boldsymbol{\mu}}\|_2^2 \cdot \sqrt{n\log(n)},$$

where $c_3$ is an absolute constant. Therefore we have

$$n\|\widetilde{\boldsymbol{\mu}}\|_2^2 - c_3\|\widetilde{\boldsymbol{\mu}}\|_2^2 \cdot \sqrt{n\log(n)} \leq \|\mathbf{Q}\boldsymbol{\mu}\|_2^2 = \|\mathbf{Z}\widetilde{\boldsymbol{\mu}}\|_2^2 \leq n\|\widetilde{\boldsymbol{\mu}}\|_2^2 + c_3\|\widetilde{\boldsymbol{\mu}}\|_2^2 \cdot \sqrt{n\log(n)}, \quad \text{(A.10)}$$

and

$$\frac{n - c_3\sqrt{n\log(n)}}{\text{tr}(\mathbf{\Sigma}) + \epsilon_\lambda} \cdot \|\widetilde{\boldsymbol{\mu}}\|_2 \leq \boldsymbol{\nu}^\top \mathbf{A}^{-1}\boldsymbol{\nu} \leq \frac{n + c_3\sqrt{n\log(n)}}{\text{tr}(\mathbf{\Sigma}) - \epsilon_\lambda} \cdot \|\widetilde{\boldsymbol{\mu}}\|_2$$

Similarly for $\mathbf{y}^\top \mathbf{A}^{-1}\boldsymbol{\nu}$, by Cauchy-Schwarz inequality, for large enough $n$ we have

$$|\mathbf{y}^\top \mathbf{A}^{-1}\boldsymbol{\nu}| = |\mathbf{y}^\top (\mathbf{Q}\mathbf{Q}^\top)^{-1}\mathbf{Q}\boldsymbol{\mu}| \leq \|\mathbf{y}\|_2 \cdot \|(\mathbf{Q}\mathbf{Q}^\top)^{-1}\mathbf{Q}\boldsymbol{\mu}\|_2 = \sqrt{n} \cdot \sqrt{\boldsymbol{\mu}^\top \mathbf{Q}^\top (\mathbf{Q}\mathbf{Q}^\top)^{-2}\mathbf{Q}\boldsymbol{\mu}}.$$

Applying Lemma A.4 and the inequality (A.10), we have

$$|\mathbf{y}^\top \mathbf{A}^{-1}\boldsymbol{\nu}| \leq \frac{\sqrt{n}}{\text{tr}(\mathbf{\Sigma}) - \epsilon_\lambda}\|\mathbf{Q}\boldsymbol{\mu}\|_2 \leq \frac{\sqrt{n} \cdot \sqrt{n + c_3\sqrt{n\log(n)}}}{\text{tr}(\mathbf{\Sigma}) - \epsilon_\lambda}\|\widetilde{\boldsymbol{\mu}}\|_2 \leq \frac{c_4 n}{\text{tr}(\mathbf{\Sigma}) - \epsilon_\lambda}\|\widetilde{\boldsymbol{\mu}}\|_2,$$

where $c_4$ is an absolute constant. Note that $\|\widetilde{\boldsymbol{\mu}}\|_2 = \|\boldsymbol{\mu}\|_{\mathbf{\Sigma}}$. This completes the proof. $\qquad\square$

# B  Proof of Theorem 3.2

Here we present the proof of Theorem 3.2.

*Proof of Theorem 3.2.* By the lower bound of the Gaussian cumulative distribution function [7], we have that for any $\boldsymbol{\theta} \in \mathbb{R}^d$,

$$R(\boldsymbol{\theta}) \geq c_1 \exp\left(-\frac{c_2(\boldsymbol{\theta}^\top \boldsymbol{\mu})^2}{\|\boldsymbol{\theta}\|_{\mathbf{\Sigma}}^2}\right), \tag{B.1}$$

where $c_1, c_2 > 0$ are absolute constants. By Proposition 4.1, we have

$$\widehat{\boldsymbol{\theta}}_{\text{SVM}} = \widehat{\boldsymbol{\theta}}_{\text{LS}} = \mathbf{X}^\top (\mathbf{X}\mathbf{X}^\top)^{-1}\mathbf{y}.$$

Plugging it into (B.1), we obtain

$$R(\widehat{\boldsymbol{\theta}}_{\text{SVM}}) \geq c_1 \exp\left\{-\frac{c_2[\mathbf{y}^\top (\mathbf{X}\mathbf{X}^\top)^{-1}\mathbf{X}\boldsymbol{\mu}]^2}{\|\mathbf{X}^\top (\mathbf{X}\mathbf{X}^\top)^{-1}\mathbf{y}\|_{\mathbf{\Sigma}}^2}\right\}. \tag{B.2}$$

Note that based on our model, we have $\mathbf{X} = \mathbf{y}\boldsymbol{\mu}^\top + \mathbf{Q}$, and

$$\begin{aligned}
\|\mathbf{X}^\top (\mathbf{X}\mathbf{X}^\top)^{-1}\mathbf{y}\|_{\mathbf{\Sigma}} &= \|(\mathbf{y}\boldsymbol{\mu}^\top + \mathbf{Q})^\top (\mathbf{X}\mathbf{X}^\top)^{-1}\mathbf{y}\|_{\mathbf{\Sigma}} \\
&\geq \left|\|\boldsymbol{\mu}\mathbf{y}^\top (\mathbf{X}\mathbf{X}^\top)^{-1}\mathbf{y}\|_{\mathbf{\Sigma}} - \|\mathbf{Q}^\top (\mathbf{X}\mathbf{X}^\top)^{-1}\mathbf{y}\|_{\mathbf{\Sigma}}\right| \\
&= \left|\mathbf{y}^\top (\mathbf{X}\mathbf{X}^\top)^{-1}\mathbf{y} \cdot \|\boldsymbol{\mu}\|_{\mathbf{\Sigma}} - \|\mathbf{Q}^\top (\mathbf{X}\mathbf{X}^\top)^{-1}\mathbf{y}\|_{\mathbf{\Sigma}}\right|
\end{aligned} \tag{B.3}$$

Plugging the above bound into (B.2), we obtain

$$R(\boldsymbol{\theta}) \geq c_1 \exp\left\{-\frac{c_2[\mathbf{y}^\top (\mathbf{X}\mathbf{X}^\top)^{-1}\mathbf{X}\boldsymbol{\mu}]^2}{(\mathbf{y}^\top (\mathbf{X}\mathbf{X}^\top)^{-1}\mathbf{y} \cdot \|\boldsymbol{\mu}\|_{\mathbf{\Sigma}} - \|\mathbf{Q}^\top (\mathbf{X}\mathbf{X}^\top)^{-1}\mathbf{y}\|_{\mathbf{\Sigma}})^2}\right\}. \tag{B.4}$$

Denote $\boldsymbol{\nu} = \mathbf{Q}\boldsymbol{\mu}$ and $\mathbf{A} = \mathbf{Q}\mathbf{Q}^\top$. Then by Lemma A.5 and the model definition $\mathbf{X} = \mathbf{y}\boldsymbol{\mu}^\top + \mathbf{Q}$, we have

$$\begin{aligned}
\mathbf{y}^\top (\mathbf{X}\mathbf{X}^\top)^{-1}\mathbf{X}\boldsymbol{\mu} &= D^{-1}[(1 + \mathbf{y}^\top \mathbf{A}^{-1}\boldsymbol{\nu})\mathbf{y}^\top \mathbf{A}^{-1} - \mathbf{y}^\top \mathbf{A}^{-1}\mathbf{y} \cdot \boldsymbol{\nu}^\top \mathbf{A}^{-1}](\mathbf{y}\boldsymbol{\mu}^\top + \mathbf{Q})\boldsymbol{\mu} \\
&= D^{-1}[(1 + \mathbf{y}^\top \mathbf{A}^{-1}\boldsymbol{\nu})\mathbf{y}^\top \mathbf{A}^{-1} - \mathbf{y}^\top \mathbf{A}^{-1}\mathbf{y} \cdot \boldsymbol{\nu}^\top \mathbf{A}^{-1}](\mathbf{y} \cdot \|\boldsymbol{\mu}\|_2^2 + \mathbf{Q}\boldsymbol{\mu}) \\
&= D^{-1}[(1 + \mathbf{y}^\top \mathbf{A}^{-1}\boldsymbol{\nu})\mathbf{y}^\top \mathbf{A}^{-1}\mathbf{y} - \mathbf{y}^\top \mathbf{A}^{-1}\mathbf{y} \cdot \boldsymbol{\nu}^\top \mathbf{A}^{-1}\mathbf{y}] \cdot \|\boldsymbol{\mu}\|_2^2 \\
&\quad + D^{-1}[(1 + \mathbf{y}^\top \mathbf{A}^{-1}\boldsymbol{\nu})\mathbf{y}^\top \mathbf{A}^{-1}\boldsymbol{\nu} - \mathbf{y}^\top \mathbf{A}^{-1}\mathbf{y} \cdot \boldsymbol{\nu}^\top \mathbf{A}^{-1}\boldsymbol{\nu}]) \\
&= D^{-1} \cdot [(\|\boldsymbol{\mu}\|_2^2 - \boldsymbol{\nu}^\top \mathbf{A}^{-1}\boldsymbol{\nu})\mathbf{y}^\top \mathbf{A}^{-1}\mathbf{y} + (1 + \mathbf{y}^\top \mathbf{A}^{-1}\boldsymbol{\nu})\mathbf{y}^\top \mathbf{A}^{-1}\boldsymbol{\nu}], \tag{B.5}
\end{aligned}$$

where the third equality follows by the notation $\boldsymbol{\nu} = \mathbf{Q}\boldsymbol{\mu}$. By Lemma A.6 and the assumption that $\text{tr}(\mathbf{\Sigma}) \geq C\max\{\epsilon_\lambda, n\|\mathbf{\Sigma}\|_2, n\|\boldsymbol{\mu}\|_{\mathbf{\Sigma}}\}$ for some large enough constant $C$, when $n$ is large enough we have

$$|\mathbf{y}^\top \mathbf{A}^{-1}\boldsymbol{\nu}| \leq \frac{c_3 n}{\text{tr}(\mathbf{\Sigma}) - \epsilon_\lambda}\|\boldsymbol{\mu}\|_{\mathbf{\Sigma}} \leq \frac{2c_4 n}{\text{tr}(\mathbf{\Sigma})}\|\boldsymbol{\mu}\|_{\mathbf{\Sigma}} \leq 1,$$

$$0 \leq \boldsymbol{\nu}^\top \mathbf{A}^{-1}\boldsymbol{\nu} \leq \frac{n + c_5\sqrt{n\log(n)}}{\text{tr}(\mathbf{\Sigma}) - \epsilon_\lambda} \cdot \|\boldsymbol{\mu}\|_{\mathbf{\Sigma}}^2 \leq \frac{2n}{\text{tr}(\mathbf{\Sigma})} \cdot \|\boldsymbol{\mu}\|_{\mathbf{\Sigma}}^2 \leq \frac{2n\|\mathbf{\Sigma}\|_2}{\text{tr}(\mathbf{\Sigma})} \cdot \|\boldsymbol{\mu}\|_2^2 \leq \frac{1}{2} \cdot \|\boldsymbol{\mu}\|_2^2,$$

$$0 \leq \mathbf{y}^\top \mathbf{A}^{-1}\mathbf{y} \leq \frac{n}{\text{tr}(\mathbf{\Sigma}) - \epsilon_\lambda} \leq \frac{2n}{\text{tr}(\mathbf{\Sigma})},$$

where $c_3, c_4$ are absolute constants. Plugging the bounds above into (A.1), we obtain

$$|\mathbf{y}^\top(\mathbf{X}\mathbf{X}^\top)^{-1}\mathbf{X}\boldsymbol{\mu}| \leq D^{-1} \cdot \left( \|\boldsymbol{\mu}\|_2^2 \cdot \mathbf{y}^\top \mathbf{A}^{-1}\mathbf{y} + 2 \cdot |\mathbf{y}^\top \mathbf{A}^{-1}\boldsymbol{\nu}| \right)$$

$$\leq D^{-1} \cdot \left[ \frac{2n}{\text{tr}(\boldsymbol{\Sigma})} \cdot \|\boldsymbol{\mu}\|_2^2 + \frac{4n}{\text{tr}(\boldsymbol{\Sigma})}\|\boldsymbol{\mu}\|_{\boldsymbol{\Sigma}} \right]$$

$$\leq D^{-1} \cdot \frac{2n}{\text{tr}(\boldsymbol{\Sigma})} \cdot (\|\boldsymbol{\mu}\|_2^2 + 2\|\boldsymbol{\mu}\|_{\boldsymbol{\Sigma}})$$

$$\leq D^{-1} \cdot \frac{4n}{\text{tr}(\boldsymbol{\Sigma})} \cdot \|\boldsymbol{\mu}\|_2^2,$$

where the last inequality follows by the assumption that $\|\boldsymbol{\mu}\|_2^2 \geq C\|\boldsymbol{\mu}\|_{\boldsymbol{\Sigma}}$ for some large enough absolute constant $C$. Therefore we have

$$[\mathbf{y}^\top(\mathbf{X}\mathbf{X}^\top)^{-1}\mathbf{X}\boldsymbol{\mu}]^2 \leq D^{-2} \cdot \frac{n^2}{64[\text{tr}(\boldsymbol{\Sigma})]^2} \cdot \|\boldsymbol{\mu}\|_2^4 = \frac{H(\boldsymbol{\mu}, \mathbf{Q}, \mathbf{y}, \boldsymbol{\Sigma})}{64} \cdot n^2\|\boldsymbol{\mu}\|_2^4, \tag{B.6}$$

where

$$H(\boldsymbol{\mu}, \mathbf{Q}, \mathbf{y}, \boldsymbol{\Sigma}) := [D \cdot \text{tr}(\boldsymbol{\Sigma})]^{-2} > 0.$$

We now proceed to study the two terms in the denominator of the exponent in (B.4). We denote

$$J_1 = \mathbf{y}^\top(\mathbf{X}\mathbf{X}^\top)^{-1}\mathbf{y} \cdot \|\boldsymbol{\mu}\|_{\boldsymbol{\Sigma}},$$
$$J_2 = \|\mathbf{Q}^\top(\mathbf{X}\mathbf{X}^\top)^{-1}\mathbf{y}\|_{\boldsymbol{\Sigma}})^2.$$

Then for $J_1$, with the same derivation as the proof of Lemma 4.4 for $I_2$, we have

$$J_1 = \sqrt{I_2} \leq 2\sqrt{H(\boldsymbol{\mu}, \mathbf{Q}, \mathbf{y}, \boldsymbol{\Sigma})} \cdot n \cdot \|\boldsymbol{\mu}\|_{\boldsymbol{\Sigma}}.$$

Moreover we also have

$$\mathbf{y}^\top(\mathbf{X}\mathbf{X}^\top)^{-1}\mathbf{y} = D^{-1} \cdot \mathbf{y}^\top \mathbf{A}^{-1}\mathbf{y} \geq D^{-1} \cdot \frac{n}{\text{tr}(\boldsymbol{\Sigma}) + \epsilon_\lambda} \geq (2D)^{-1} \cdot \frac{n}{\text{tr}(\boldsymbol{\Sigma})},$$

where the first inequality follows by Lemma A.6, and the second inequality follows by the assumption that $\text{tr}(\boldsymbol{\Sigma}) \geq C\epsilon_\lambda$ for some large enough constant $C$. Then we have

$$J_1 = \mathbf{y}^\top(\mathbf{X}\mathbf{X}^\top)^{-1}\mathbf{y} \cdot \|\boldsymbol{\mu}\|_{\boldsymbol{\Sigma}} \geq (2D)^{-1} \cdot \frac{n \cdot \|\boldsymbol{\mu}\|_{\boldsymbol{\Sigma}}}{\text{tr}(\boldsymbol{\Sigma})} = (1/2) \cdot \sqrt{H(\boldsymbol{\mu}, \mathbf{Q}, \mathbf{y}, \boldsymbol{\Sigma})} \cdot n \cdot \|\boldsymbol{\mu}\|_{\boldsymbol{\Sigma}},$$

where we use the definition $H(\boldsymbol{\mu}, \mathbf{Q}, \mathbf{y}, \boldsymbol{\Sigma}) = [D \cdot \text{tr}(\boldsymbol{\Sigma})]^{-2}$. Therefore in summary we have

$$(1/2) \cdot \sqrt{H(\boldsymbol{\mu}, \mathbf{Q}, \mathbf{y}, \boldsymbol{\Sigma})} \cdot n \cdot \|\boldsymbol{\mu}\|_{\boldsymbol{\Sigma}} \leq J_1 \leq 2\sqrt{H(\boldsymbol{\mu}, \mathbf{Q}, \mathbf{y}, \boldsymbol{\Sigma})} \cdot n \cdot \|\boldsymbol{\mu}\|_{\boldsymbol{\Sigma}}, \tag{B.7}$$

where $c_5$ is an absolute constant. Similarly, for $J_2$, with the same derivation as the proof of Lemma 4.4 for $I_3$, we have

$$J_2^2 = I_3 \leq c_5 H(\boldsymbol{\mu}, \mathbf{Q}, \mathbf{y}, \boldsymbol{\Sigma}) \cdot (n \cdot \|\boldsymbol{\Sigma}\|_F^2 + n^2 \cdot \|\boldsymbol{\Sigma}\|_2^2). \tag{B.8}$$

Moreover, we denote $\mathbf{a} = D^{-1}[(1 + \mathbf{y}^\top \mathbf{A}^{-1}\boldsymbol{\nu}) \cdot \mathbf{y} - \mathbf{y}^\top \mathbf{A}^{-1}\mathbf{y} \cdot \boldsymbol{\nu}]$. Then with the same derivation,

$$J_2^2 = \mathbf{y}^\top(\mathbf{X}\mathbf{X}^\top)^{-1}\mathbf{Q}\boldsymbol{\Sigma}\mathbf{Q}^\top(\mathbf{X}\mathbf{X}^\top)^{-1}\mathbf{y}$$
$$= \mathbf{a}^\top(\mathbf{Q}\mathbf{Q}^\top)^{-1}\mathbf{Q}\boldsymbol{\Sigma}\mathbf{Q}^\top(\mathbf{Q}\mathbf{Q}^\top)^{-1}\mathbf{a}$$
$$= \mathbf{a}^\top(\mathbf{Z}\boldsymbol{\Lambda}\mathbf{Z}^\top)^{-1}\mathbf{Z}\boldsymbol{\Lambda}^2\mathbf{Z}^\top(\mathbf{Z}\boldsymbol{\Lambda}\mathbf{Z}^\top)^{-1}\mathbf{a}, \tag{B.9}$$

where we plug in $\boldsymbol{\Sigma} = \mathbf{V}\boldsymbol{\Lambda}\mathbf{V}^\top$ and $\mathbf{Q} = \mathbf{Z}\boldsymbol{\Lambda}^{1/2}\mathbf{V}^\top$ for $\mathbf{Z}$ with independent sub-Gaussian entries. We have

$$J_2^2 = \mathbf{a}^\top(\mathbf{Z}\boldsymbol{\Lambda}\mathbf{Z}^\top)^{-1}\mathbf{Z}\boldsymbol{\Lambda}^2\mathbf{Z}^\top(\mathbf{Z}\boldsymbol{\Lambda}\mathbf{Z}^\top)^{-1}\mathbf{a}$$
$$\geq \mathbf{a}^\top(\mathbf{Z}\boldsymbol{\Lambda}\mathbf{Z}^\top)^{-2}\mathbf{a} \cdot \left[ \|\boldsymbol{\Sigma}\|_F^2 - \epsilon_\lambda' \right]$$
$$\geq \|\mathbf{a}\|_2^2 \cdot \frac{\|\boldsymbol{\Sigma}\|_F^2 - \epsilon_\lambda'}{[\text{tr}(\boldsymbol{\Sigma}) + \epsilon_\lambda]^2}$$
$$\geq \|\mathbf{a}\|_2^2 \cdot \frac{\|\boldsymbol{\Sigma}\|_F^2 - \epsilon_\lambda'}{2[\text{tr}(\boldsymbol{\Sigma})]^2}. \tag{B.10}$$

Here the first inequality follows by Lemma A.7, the second inequality follows by Lemma A.4, and the third inequality follows by the assumption that $\text{tr}(\boldsymbol{\Sigma}) \geq C\epsilon_\lambda$ for some large enough absolute constant $C$. By the definition of $\epsilon_\lambda'$ in Lemma A.7 and Cauchy-Schwarz inequality, we have

$$
\begin{aligned}
\epsilon_\lambda' &:= c_6\big(n \cdot \|\boldsymbol{\Sigma}\|_2^2 + \sqrt{n} \cdot \|\boldsymbol{\Sigma}^2\|_F\big) \\
&\leq c_6\big(n \cdot \|\boldsymbol{\Sigma}\|_2^2 + \sqrt{n} \cdot \|\boldsymbol{\Sigma}\|_2 \cdot \|\boldsymbol{\Sigma}\|_F\big) \\
&\leq c_6\big(n \cdot \|\boldsymbol{\Sigma}\|_2^2 + 2c_6 n \cdot \|\boldsymbol{\Sigma}\|_2^2 + \|\boldsymbol{\Sigma}\|_F^2/(2c_6)\big) \\
&\leq c_7 n \cdot \|\boldsymbol{\Sigma}\|_2^2 + \|\boldsymbol{\Sigma}\|_F^2/2,
\end{aligned}
$$

where $c_6, c_7$ are absolute constants. Plugging the above bound into (B.10) gives

$$
J_2^2 \geq \|\mathbf{a}\|_2^2 \cdot \frac{\|\boldsymbol{\Sigma}\|_F^2 - c_8 n \cdot \|\boldsymbol{\Sigma}\|_2^2}{4[\text{tr}(\boldsymbol{\Sigma})]^2} \tag{B.11}
$$

for some absolute constant $c_8$. Moreover, by the definition $\mathbf{a}$ and the triangle inequality, we have

$$
\begin{aligned}
\|\mathbf{a}\|_2^2 &= \|D^{-1}(1 + \mathbf{y}^\top \mathbf{A}^{-1}\boldsymbol{\nu})\mathbf{y} - \mathbf{y}^\top \mathbf{A}^{-1}\mathbf{y} \cdot \boldsymbol{\nu}\|_2^2 \\
&\geq \big[D^{-1}(1 + \mathbf{y}^\top \mathbf{A}^{-1}\boldsymbol{\nu})\|\mathbf{y}\|_2 - D^{-1}(\mathbf{y}^\top \mathbf{A}^{-1}\mathbf{y}) \cdot \|\mathbf{Q}\boldsymbol{\mu}\|_2\big]^2 \\
&= D^{-2}\big[(1 + \mathbf{y}^\top \mathbf{A}^{-1}\boldsymbol{\nu}) \cdot \sqrt{n} - (\mathbf{y}^\top \mathbf{A}^{-1}\mathbf{y}) \cdot \|\mathbf{Q}\boldsymbol{\mu}\|_2\big]^2.
\end{aligned} \tag{B.12}
$$

By Lemma A.6 and the assumption that $\text{tr}(\boldsymbol{\Sigma}) \geq C\max\{\epsilon_\lambda, n, n\|\boldsymbol{\mu}\|_{\boldsymbol{\Sigma}}\}$ for some large enough constant $C$, we have

$$
\begin{aligned}
|\mathbf{y}^\top \mathbf{A}^{-1}\boldsymbol{\nu}| &\leq \frac{c_9 n}{\text{tr}(\boldsymbol{\Sigma}) - \epsilon_\lambda}\|\boldsymbol{\mu}\|_{\boldsymbol{\Sigma}} \leq 1/2, \\
\mathbf{y}^\top \mathbf{A}^{-1}\mathbf{y} &\leq \frac{n}{\text{tr}(\boldsymbol{\Sigma}) - \epsilon_\lambda} \leq \frac{2n}{\text{tr}(\boldsymbol{\Sigma})},
\end{aligned}
$$

where $c_9$ is an absolute constant. Moreover, with the same proof as in Lemma A.6, when $n$ is sufficiently large, with probability at least $1 - O(n^{-2})$ we have

$$
\|\mathbf{Q}\boldsymbol{\mu}\|_2^2 \leq 2n\|\boldsymbol{\mu}\|_{\boldsymbol{\Sigma}}^2.
$$

Utilizing these inequalities above, we have

$$
\begin{aligned}
(1 + \mathbf{y}^\top \mathbf{A}^{-1}\boldsymbol{\nu}) \cdot \sqrt{n} &\geq \sqrt{n}/2, \\
(\mathbf{y}^\top \mathbf{A}^{-1}\mathbf{y}) \cdot \|\mathbf{Q}\boldsymbol{\mu}\|_2 &\leq \frac{2n}{\text{tr}(\boldsymbol{\Sigma})} \cdot \sqrt{2n}\|\boldsymbol{\mu}\|_{\boldsymbol{\Sigma}} \leq \sqrt{n}/4,
\end{aligned}
$$

where the second line above follows the assumption that $\text{tr}(\boldsymbol{\Sigma}) \geq Cn\|\boldsymbol{\mu}\|_{\boldsymbol{\Sigma}}$ for some large enough constant $C$. Combining these bounds with (B.12), we have

$$
\|\mathbf{a}\|_2^2 \geq D^{-2}\big[(1 + \mathbf{y}^\top \mathbf{A}^{-1}\boldsymbol{\nu}) \cdot \sqrt{n} - (\mathbf{y}^\top \mathbf{A}^{-1}\mathbf{y}) \cdot \|\mathbf{Q}\boldsymbol{\mu}\|_2\big]^2 \geq D^{-2}n/16.
$$

Further plugging this bound into (B.11), we have

$$
J_2^2 \geq \frac{n}{16D^2} \cdot \frac{\|\boldsymbol{\Sigma}\|_F^2 - c_8 n \cdot \|\boldsymbol{\Sigma}\|_2^2}{4[\text{tr}(\boldsymbol{\Sigma})]^2} = H(\boldsymbol{\mu}, \mathbf{Q}, \mathbf{y}, \boldsymbol{\Sigma}) \cdot (c_{10} n \cdot \|\boldsymbol{\Sigma}\|_F^2 - c_{11} n^2 \cdot \|\boldsymbol{\Sigma}\|_2^2), \quad \text{(B.13)}
$$

where $c_{10}, c_{11}$ are absolute constants, and we use the definition $H(\boldsymbol{\mu}, \mathbf{Q}, \mathbf{y}, \boldsymbol{\Sigma}) = [D \cdot \text{tr}(\boldsymbol{\Sigma})]^{-2}$. Combining (B.8) and (B.13), we obtain

$$
H(\boldsymbol{\mu}, \mathbf{Q}, \mathbf{y}, \boldsymbol{\Sigma}) \cdot (c_{10} n\|\boldsymbol{\Sigma}\|_F^2 - c_{11} n^2\|\boldsymbol{\Sigma}\|_2^2) \leq J_2^2 \leq c_5 H(\boldsymbol{\mu}, \mathbf{Q}, \mathbf{y}, \boldsymbol{\Sigma}) \cdot (n\|\boldsymbol{\Sigma}\|_F^2 + n^2\|\boldsymbol{\Sigma}\|_2^2). \tag{B.14}
$$

In the rest of the proof, we consider the two cases in Theorem 3.2 separately based on (B.7) and (B.14).

**Case 1.** Suppose that $n\|\boldsymbol{\mu}\|_{\boldsymbol{\Sigma}}^2 \geq C(\|\boldsymbol{\Sigma}\|_F^2 + n\|\boldsymbol{\Sigma}\|_2^2)$ for some large enough constant $C$. Then by (B.7) and (B.14), we have

$$
J_1 \geq (1/2) \cdot \sqrt{H(\boldsymbol{\mu}, \mathbf{Q}, \mathbf{y}, \boldsymbol{\Sigma})} \cdot n \cdot \|\boldsymbol{\mu}\|_{\boldsymbol{\Sigma}}
$$

$$
J_2 \leq 2\sqrt{c_5}\sqrt{H(\boldsymbol{\mu}, \mathbf{Q}, \mathbf{y}, \boldsymbol{\Sigma})} \cdot \sqrt{n \cdot \|\boldsymbol{\Sigma}\|_F^2 + n^2 \cdot \|\boldsymbol{\Sigma}\|_2^2} \leq (1/4) \cdot \sqrt{H(\boldsymbol{\mu}, \mathbf{Q}, \mathbf{y}, \boldsymbol{\Sigma})} \cdot n \cdot \|\boldsymbol{\mu}\|_{\boldsymbol{\Sigma}}
$$

Plugging the above inequalities and (B.6) into (B.4), we obtain Therefore by

$$R(\boldsymbol{\theta}) \geq c_1 \exp\left\{ - \frac{c_2 n^2 \|\boldsymbol{\mu}\|_2^4/64}{(n \cdot \|\boldsymbol{\mu}\|_{\boldsymbol{\Sigma}}/4)^2} \right\} = c_1 \exp\left\{ - \frac{c_{12}\|\boldsymbol{\mu}\|_2^4}{\|\boldsymbol{\mu}\|_{\boldsymbol{\Sigma}}^2} \right\},$$

where $c_{12}$ is an absolute constant. This completes the proof of the first case in Theorem 3.2.

**Case 2.** Suppose that $\|\boldsymbol{\Sigma}\|_F^2 \geq Cn(\|\boldsymbol{\mu}\|_{\boldsymbol{\Sigma}}^2 + \|\boldsymbol{\Sigma}\|_2^2)$ for some large enough constant $C$. Then by (B.7) we have

$$J_1 \leq 2\sqrt{H(\boldsymbol{\mu}, \mathbf{Q}, \mathbf{y}, \boldsymbol{\Sigma})} \cdot n \cdot \|\boldsymbol{\mu}\|_{\boldsymbol{\Sigma}} \leq \sqrt{H(\boldsymbol{\mu}, \mathbf{Q}, \mathbf{y}, \boldsymbol{\Sigma})} \cdot \sqrt{c_{10}}\|\boldsymbol{\Sigma}\|_F/4. \qquad (\text{B.15})$$

Moreover for $J_2$, by (B.14) we have

$$J_2^2 \geq H(\boldsymbol{\mu}, \mathbf{Q}, \mathbf{y}, \boldsymbol{\Sigma}) \cdot (c_{10}n\|\boldsymbol{\Sigma}\|_F^2 - c_{11}n^2\|\boldsymbol{\Sigma}\|_2^2) \geq H(\boldsymbol{\mu}, \mathbf{Q}, \mathbf{y}, \boldsymbol{\Sigma}) \cdot c_{10}n\|\boldsymbol{\Sigma}\|_F^2/4,$$

and therefore

$$J_2 \geq \sqrt{H(\boldsymbol{\mu}, \mathbf{Q}, \mathbf{y}, \boldsymbol{\Sigma})} \cdot \sqrt{c_{10}n}\|\boldsymbol{\Sigma}\|_F/2. \qquad (\text{B.16})$$

Plugging (B.6), (B.15) and (B.16) into (B.4), we obtain

$$R(\boldsymbol{\theta}) \geq c_1 \exp\left\{ - \frac{c_2 n^2 \|\boldsymbol{\mu}\|_2^4/64}{(\sqrt{c_{10}n}\|\boldsymbol{\Sigma}\|_F/4)^2} \right\} = c_1 \exp\left\{ - \frac{c_{13}n\|\boldsymbol{\mu}\|_2^4}{\|\boldsymbol{\Sigma}\|_F^2} \right\},$$

where $c_{13}$ is an absolute constant. This completes the proof of the second case in Theorem 3.2. $\qquad \square$

## C  Proof of Corollaries

Here we provide the proof of the Corollaries 3.3, 3.5 and 3.7 in Section 3.

### C.1  Proof of Corollary 3.3

The proof of Corollary 3.3 is a direct application of Theorem 3.1. The detailed proof is as follows.

*Proof of Corollary 3.3.* When $\boldsymbol{\Sigma} = \mathbf{I}$, we have $\text{tr}(\boldsymbol{\Sigma}) = d$, $\|\boldsymbol{\Sigma}\|_2 = 1$, $\|\boldsymbol{\Sigma}\|_F = \sqrt{d}$ and $\|\boldsymbol{\mu}\|_{\boldsymbol{\Sigma}} = \|\boldsymbol{\mu}\|_2$. Under the condition in Corollary 3.3 that $d \geq C \max\left\{ n^2, n\sqrt{\log(n)} \cdot \|\boldsymbol{\mu}\|_2 \right\}$ and $\|\boldsymbol{\mu}\|_2 \geq C$ for some large enough absolute constant $C$, it is easy to check that the conditions of Theorem 3.1

$$\text{tr}(\boldsymbol{\Sigma}) = \Omega\big( \max\left\{ n^{3/2}\|\boldsymbol{\Sigma}\|_2, n\|\boldsymbol{\Sigma}\|_F, n\sqrt{\log(n)} \cdot \|\boldsymbol{\mu}\|_{\boldsymbol{\Sigma}} \right\}\big), \quad \|\boldsymbol{\mu}\|_2 \geq C\|\boldsymbol{\Sigma}\|_2$$

hold. Therefore by Theorem 3.1, we have

$$R(\widehat{\boldsymbol{\theta}}_{\text{SVM}}) \leq \exp\left( \frac{-c_1 n\|\boldsymbol{\mu}\|_2^4}{n\|\boldsymbol{\mu}\|_{\boldsymbol{\Sigma}}^2 + \|\boldsymbol{\Sigma}\|_F^2 + n\|\boldsymbol{\Sigma}\|_2^2} \right) \leq \exp\left( \frac{-c_2 n\|\boldsymbol{\mu}\|_2^4}{n\|\boldsymbol{\mu}\|_2^2 + d} \right),$$

where $c_1$, $c_2$ are absolute constants. This completes the proof. $\qquad \square$

### C.2  Proof of Corollary 3.5

Here we present the proof of Corollary 3.5, which is mostly based on the estimation of the order of the summations $\sum_{k=1}^{d} k^{-\alpha}$ and $\sum_{k=1}^{d} k^{-2\alpha}$. We first present the full version of the corollary with detailed dependency in the sample size $n$ as follows.

**Corollary C.1.** [Full version of Corollary 3.5] Suppose that $\lambda_k = k^{-\alpha}$, and one of the following conditions hold:

1. $\alpha \in [0, 1/2)$, $d = \widetilde{\Omega}(n^{\frac{3}{2(1-\alpha)}} + n^2 + (n\|\boldsymbol{\mu}\|_{\boldsymbol{\Sigma}})^{\frac{1}{1-\alpha}})$, and $\|\boldsymbol{\mu}\|_2 = \omega(1 + n^{-1/4}d^{1/4-\alpha/2})$.

2. $\alpha = 1/2$, $d = \widetilde{\Omega}(n^3 + n^2\|\boldsymbol{\mu}\|_{\boldsymbol{\Sigma}}^2)$, and $\|\boldsymbol{\mu}\|_2 = \omega(1 + n^{-1/4}(\log(d))^{1/4})$.

3. $\alpha \in (1/2, 1)$, $d = \widetilde{\Omega}(n^{\frac{3}{2(1-\alpha)}} + (n\|\boldsymbol{\mu}\|_{\boldsymbol{\Sigma}})^{\frac{1}{1-\alpha}})$, and $\|\boldsymbol{\mu}\|_2 = \omega(1)$.

Then with probability at least $1 - n^{-1}$, the population risk of the maximum margin classifier satisfies $R(\widehat{\boldsymbol{\theta}}_{\text{SVM}}) = o(1)$.

*Proof of Corollary C.1.* We first consider the case when $\alpha \in [0, 1/2)$. We have

$$\text{tr}(\mathbf{\Sigma}) = \sum_{k=1}^d \lambda_k = \sum_{k=1}^d k^{-\alpha} \geq \int_{t=1}^d t^{-\alpha} \mathrm{d}t = \frac{d^{1-\alpha}}{1-\alpha} - \frac{1}{1-\alpha} > \frac{d^{1-\alpha}}{2(1-\alpha)}$$

when $d$ is sufficiently large. Similarly, we have

$$\|\mathbf{\Sigma}\|_F^2 = \sum_{k=1}^d \lambda_k^2 = 1 + \sum_{k=2}^d k^{-2\alpha} \leq 1 + \int_{t=1}^{d-1} t^{-2\alpha} \mathrm{d}t = 1 + \frac{(d-1)^{1-2\alpha}}{1-2\alpha} - \frac{1}{1-2\alpha} \leq 1 + \frac{d^{1-2\alpha}}{1-2\alpha}.$$

Therefore, a sufficient condition for the assumptions in Theorem 3.1 to hold is that $\|\boldsymbol{\mu}\|_2 = \omega(1)$ and

$$\frac{d^{1-\alpha}}{2(1-\alpha)} \geq Cn^{3/2},$$

$$\frac{d^{1-\alpha}}{2(1-\alpha)} \geq Cn \cdot \sqrt{1 + \frac{d^{1-2\alpha}}{1-2\alpha}},$$

$$\frac{d^{1-\alpha}}{2(1-\alpha)} \geq Cn\sqrt{\log(n)} \cdot \|\boldsymbol{\mu}\|_{\mathbf{\Sigma}}.$$

After simplifying the result, we derive the condition that $d = \widetilde{\Omega}(n^{\frac{3}{2(1-\alpha)}} + n^2 + (n\|\boldsymbol{\mu}\|_{\mathbf{\Sigma}})^{\frac{1}{1-\alpha}})$. We further check the conditions on $\|\boldsymbol{\mu}\|_2$ that lead to $o(1)$ population risk. Note that when $\|\boldsymbol{\mu}\|_2 = \omega(1)$, $\|\boldsymbol{\mu}\|_2^4/\|\boldsymbol{\mu}\|_{\mathbf{\Sigma}}^2 = \omega(1)$. We also check the condition that $n\|\boldsymbol{\mu}\|_2^4/\|\mathbf{\Sigma}\|_F^2 = \omega(1)$. A sufficient condition is that

$$n\|\boldsymbol{\mu}\|_2^4 = \omega\left(1 + \frac{d^{1-2\alpha}}{1-2\alpha}\right).$$

Simplifying the condition completes the proof for the case $\alpha \in [0, 1/2)$.

For the case $\alpha = 1/2$, we have

$$\text{tr}(\mathbf{\Sigma}) = \sum_{k=1}^d \lambda_k = \sum_{k=1}^d k^{-1/2} \geq \int_{t=1}^d t^{-1/2} \mathrm{d}t = \frac{d^{1-1/2}}{1-1/2} - \frac{1}{1-1/2} > \sqrt{d}$$

when $d$ is sufficiently large. Moreover,

$$\|\mathbf{\Sigma}\|_F^2 = \sum_{k=1}^d \lambda_k^2 = 1 + \sum_{k=2}^d k^{-1} \leq 1 + \int_{t=1}^{d-1} t^{-1} \mathrm{d}t = 1 + \log(d-1) \leq 1 + \log(d).$$

Verifying the conditions

$$\sqrt{d} \geq Cn^{3/2},$$
$$\sqrt{d} \geq Cn \cdot \sqrt{1 + \log(d)},$$
$$\sqrt{d} \geq Cn\sqrt{\log(n)} \cdot \|\boldsymbol{\mu}\|_{\mathbf{\Sigma}}$$

then gives a sufficient condition $d = \widetilde{\Omega}(n^3 + n^2\|\boldsymbol{\mu}\|_{\mathbf{\Sigma}}^2)$, $\|\boldsymbol{\mu}\|_2 = \omega(1)$ for the assumptions in Theorem 3.1 to hold. It is also easy to verify that when $\|\boldsymbol{\mu}\|_2 = \omega(1 + n^{-1/4}(\log(d))^{1/4})$ we have $R(\widehat{\boldsymbol{\theta}}_{\text{SVM}}) = o(1)$.

Finally for the case $\alpha \in (1/2, 1)$, we have

$$\text{tr}(\mathbf{\Sigma}) = \sum_{k=1}^d \lambda_k = \sum_{k=1}^d k^{-\alpha} \geq \int_{t=1}^d t^{-\alpha} \mathrm{d}t = \frac{d^{1-\alpha}}{1-\alpha} - \frac{1}{1-\alpha}.$$

Moreover, in this setting we have $\|\mathbf{\Sigma}\|_F^2 \leq c_1$ for some absolute constant $c_1$. It is therefore easy to check that $\|\boldsymbol{\mu}\|_2 = \omega(1)$ and

$$d = \widetilde{\Omega}(n^{\frac{3}{2(1-\alpha)}} + (n\|\boldsymbol{\mu}\|_{\mathbf{\Sigma}})^{\frac{1}{1-\alpha}})$$

are sufficient for the assumptions in Theorem 3.1 to hold, and we also have $R(\widehat{\boldsymbol{\theta}}_{\text{SVM}}) = o(1)$. $\quad\square$

## C.3 Proof of Corollary 3.7

The proof of Corollary 3.7 for the rare/weak feature model is rather straightforward.

*Proof of Corollary 3.7.* Note that in the rare/weak feature model we have $\|\boldsymbol{\mu}\|_2 = \gamma\sqrt{s}$. Therefore the conditions of Corollary 3.3 are satisfied and we have

$$R(\widehat{\boldsymbol{\theta}}_{\mathrm{SVM}}) \leq \exp\left(-\frac{c_1 n\|\boldsymbol{\mu}\|_2^4}{n\|\boldsymbol{\mu}\|_2^2 + d}\right) = \exp\left(-\frac{c_1 n\gamma^4 s^2}{n\gamma^2 s + d}\right),$$

where $c_1$ is an absolute constant. This completes the proof. $\qquad\square$

## D Experiments

In this section we present simulation results to backup our population risk bound in Theorem 3.1. We generate $\mathbf{u}$ as a standard Gaussian vector, and set $\boldsymbol{\Sigma} = \mathrm{diag}\{\lambda_1, \ldots, \lambda_d\}$ with $\lambda_k = k^{-\alpha}$ for some parameter $\alpha \in [0, 1)$, which matches the setting studied in Section 3. The mean vector $\boldsymbol{\mu}$ is generated uniformly from the sphere centered at the origin with radius $r$. All population risks are calculated by taking the average of 100 independent experiments. Note that under our setting, $\widehat{\boldsymbol{\theta}}_{\mathrm{SVM}} = \widehat{\boldsymbol{\theta}}_{\mathrm{LS}}$ can be easily calculated. Moreover, since we are considering Gaussian mixtures in our experiments, the population risk can be directly calculated with the Gaussian cumulative distribution function:

$$R(\widehat{\boldsymbol{\theta}}_{\mathrm{SVM}}) = \mathbb{P}[\boldsymbol{\theta}^\top\boldsymbol{\mu} < y \cdot \widehat{\boldsymbol{\theta}}_{\mathrm{SVM}}^\top\boldsymbol{\Lambda}^{1/2}\mathbf{u}].$$

The derivation of the above result is in the proof of Lemma 4.2 in Appendix A.2.1.

**Population risk versus the norm of the mean vector $\|\boldsymbol{\mu}\|_2$.** We first present experimental results on the relation between the population risk and the norm of the mean vector $\|\boldsymbol{\mu}\|_2$. Note that in our setting, the risk bound in Theorem 3.1 reduces to the following bound:

$$R(\widehat{\boldsymbol{\theta}}_{\mathrm{SVM}}) \leq \exp\left(\frac{-C'n\|\boldsymbol{\mu}\|_2^4}{n\|\boldsymbol{\mu}\|_{\boldsymbol{\Sigma}}^2 + \sum_{k=1}^d k^{-2\alpha}}\right).$$

Based on this bound, we can first see that the population risk should be smaller when $\alpha$ is larger. Moreover, the dependency of $R(\widehat{\boldsymbol{\theta}}_{\mathrm{SVM}})$ depends on the comparison between the scaling of the two terms in the denominator. When

$$\sum_{k=1}^d k^{-2\alpha} \geq n\|\boldsymbol{\mu}\|_{\boldsymbol{\Sigma}}^2, \tag{D.1}$$

we can expect that $-\log(R(\widehat{\boldsymbol{\theta}}_{\mathrm{SVM}}))$ should be roughly of order $\|\boldsymbol{\mu}\|_2^4$. On the other hand, if (D.1) does not hold, then $-\log(R(\widehat{\boldsymbol{\theta}}_{\mathrm{SVM}}))$ should be roughly of order $\|\boldsymbol{\mu}\|_2^2$. It is also clear that whether (D.1) holds heavily depends on the values of the sample size $n$ and $\alpha$: when $n$ is large, then (D.1) is less likely to be satisfied. Moreover, when $\alpha > 1/2$, (D.1) cannot hold because in this case $\sum_{k=1}^d k^{-2\alpha}$ is upper bounded by a constant.

In Figure 2, we verify the above argument by verifying the dependency of the population risk $R(\widehat{\boldsymbol{\theta}}_{\mathrm{SVM}})$ on the norm of the mean vector $\|\boldsymbol{\mu}\|_2$ with different values of $\alpha$ and sample size $n$. From Figures 2(a) and 2(c), we can see that $R(\widehat{\boldsymbol{\theta}}_{\mathrm{SVM}})$ decreases with $\|\boldsymbol{\mu}\|_2$ and $\alpha$. From 2(b), we verify that when $n = 10$ (which is rather small) and when $\alpha = 0, 0.2, 0.4$, $-\log(R(\widehat{\boldsymbol{\theta}}_{\mathrm{SVM}}))$ is linear in $\|\boldsymbol{\mu}\|_2^2$. This verifies our discussion for the setting when (D.1) holds. On the other hand, when $\alpha = 0.6, 0.8$, $-\log(R(\widehat{\boldsymbol{\theta}}_{\mathrm{SVM}}))$ has a higher order dependency in $\|\boldsymbol{\mu}\|_2^2$, which is because $\sum_{k=1}^d k^{-2\alpha}$ is upper bounded by a constant and (D.1) cannot hold. In Figure 2(d), we further verify that when $n = 100$, (D.1) never hold and $-\log(R(\widehat{\boldsymbol{\theta}}_{\mathrm{SVM}}))$ is of order $\|\boldsymbol{\mu}\|_2^2$ for all choices of $\alpha$. This set of experiments verifies our risk bound in Theorem 3.1.

**Verification of the dimension-dependent and dimension-free settings.** In Corollary 3.5, we have discussed that when $\alpha < 1/2$, achieving a small population risk requires a larger $\|\boldsymbol{\mu}\|_2$ when $d$ is larger. On the other hand, when $\alpha > 1/2$, the requirement on $\|\boldsymbol{\mu}\|_2$ to achieve small population error is dimension-free. Here we present experimental results to verify our claim. The results are given

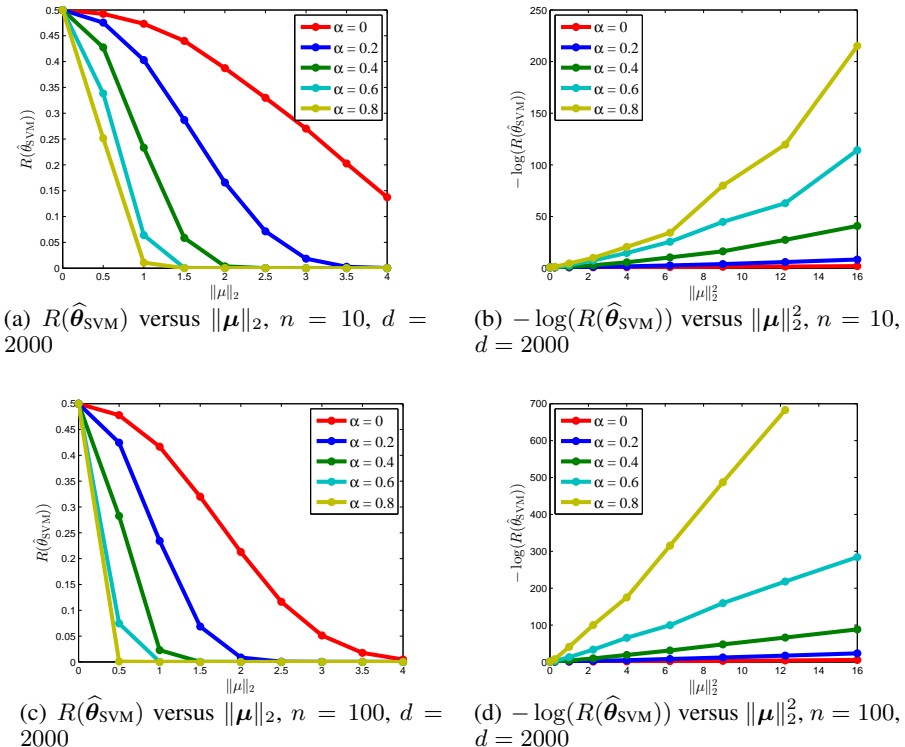

(a) $R(\widehat{\boldsymbol{\theta}}_{\mathrm{SVM}})$ versus $\|\boldsymbol{\mu}\|_2$, $n = 10$, $d = 2000$

(b) $-\log(R(\widehat{\boldsymbol{\theta}}_{\mathrm{SVM}}))$ versus $\|\boldsymbol{\mu}\|_2^2$, $n = 10$, $d = 2000$

(c) $R(\widehat{\boldsymbol{\theta}}_{\mathrm{SVM}})$ versus $\|\boldsymbol{\mu}\|_2$, $n = 100$, $d = 2000$

(d) $-\log(R(\widehat{\boldsymbol{\theta}}_{\mathrm{SVM}}))$ versus $\|\boldsymbol{\mu}\|_2^2$, $n = 100$, $d = 2000$

Figure 2: Experiments on the dependency of the population risk $R(\widehat{\boldsymbol{\theta}}_{\mathrm{SVM}})$ on the norm of the mean vector $\|\boldsymbol{\mu}\|_2$ with different values of $\alpha$ and sample size $n$. (a) and (b) gives the curves with $n = 10$, while (c) and (d) are for the case $n = 100$. Moreover, (a) and (c) gives the curves of $R(\widehat{\boldsymbol{\theta}}_{\mathrm{SVM}})$ versus $\|\boldsymbol{\mu}\|_2$, and to further test the tightness of our risk bound, in (c) and (d) we also study the relation between $-\log(R(\widehat{\boldsymbol{\theta}}_{\mathrm{SVM}}))$ and $\|\boldsymbol{\mu}\|_2^2$. The dimension $d$ is set to 2000 in all these figures. In (d) we omit the last point $\|\boldsymbol{\mu}\|_2 = 16$ in the curve for $\alpha = 0.8$ because the population risk in this case is too small and is dominated by the numerical accuracy.

in Figure 3. We can see very clearly that when $\alpha = 0.2$, the risk curves for different $d$ are different, and larger $d$ results in worse population risk. However, when $\alpha = 0.8$, all the risk curves are almost exactly the same, which indicates that the population risk is dimension-free. This verifies our claim in Corollary 3.5.

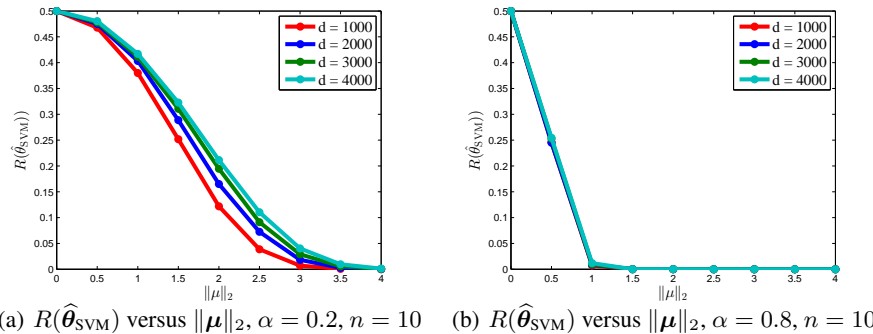

(a) $R(\widehat{\boldsymbol{\theta}}_{\mathrm{SVM}})$ versus $\|\boldsymbol{\mu}\|_2$, $\alpha = 0.2$, $n = 10$

(b) $R(\widehat{\boldsymbol{\theta}}_{\mathrm{SVM}})$ versus $\|\boldsymbol{\mu}\|_2$, $\alpha = 0.8$, $n = 10$

Figure 3: The population risk curve with respect to $\|\boldsymbol{\mu}\|_2$ with different values of $\alpha$ and dimension $d$. (a) shows the result for $\alpha = 0.2$, while (b) is for the case $\alpha = 0.8$. The sample size $n$ is set to 10 in both experiments.