# OpenReview forum: "Risk Bounds for Over-parameterized Maximum Margin Classification on Sub-Gaussian Mixtures"
_NeurIPS.cc/2021/Conference — NeurIPS 2021 Poster_

### Official Review · Reviewer_1ykL · 2021-07-13

**Rating:** 9
**Confidence:** 3

**Summary:**

The paper studies maximum-margin classifiers in the setting where the dimension of the data is much larger than the number of samples. This data is assumed to come from a mixture of two subgaussians under a standard generative model. Notably, prior work shows that max-margin classifiers (i.e. Support Vector Machines) return the same estimator as the minimum-norm solution to an underconstrained least squares problem. The research mostly pertains to analyzing that least squares solution.

The paper focuses on providing sharper guarantees on when exactly the SVM is equivalent to the LS solution, and when this LS solution has low risk/misclassification probability. Their guarantees depend exclusively on the sub-gaussian data's true covariance matrix, the distance between the two subgaussian modes, and the number of samples. Notably, this does *not* depend explicitly on the dimension of the data. So, we can better understand how the spectrum of the covariance matrix impacts the misclassification probability.

Some complementary lower bounds are given.

**Limitations And Societal Impact:**

The limitations discussed are fine and sufficient. Admittedly, they are not particularly creative or interesting.

Societal Impact: Not applicable.

**Main Review:**

### Overall Impression
This paper was a pleasure to read, and I confidently recommend it for publication.
Further, I verified much of the math in the appendix and found no errors.

### Detailed Review

The setting of the paper is interesting. The focus on characterizing as-much-as-possible in terms of the covariance matrix is solid and pays off well. The structure of the paper is very clear and easy to read, providing intuitive interpretations of the core theorems along with a high-level overview of the proof.

The background information in the introduction is also well presented. While I am familiar with all the technical tools used in this paper, I had not read papers on benign overfitting in the past. Nevertheless, I was able to quickly understand the core ideas and felt comfortable reading this paper.

I do have some questions and feedback for the authors:
1. How fundamental is the covariance matrix's trace lower bound assumption(s)? You mention that the core theorems only assume the trace is large enough, but that the risk bound itself does not depend directly on this trace. Do we have reason to think this trace assumption is fundamental or necessary? Of all metrics, why is trace meaningful here?
1. Intuitively parse the hard constraints on the trace of the covariance matrix in Theorem 3.1 [Line 167]. What kinds of covariance matrices fail this check? Do they have a flat spectrum? Does the sample complexity $n$ limit how flat the spectrum can be? [Lines 209 - 233] get close to discussing this, but fall short of explaining what covariance matrices / sub-gaussian structures fail this test, and how that depends on $n$.
    - For example, Theorem 3.1 requires $n \leq \frac{tr(\Sigma)}{\\|\Sigma\\|_F}$ amongst other constraints. Is this a reasonable way to interpret the hard constraint? What does this really say about $\Sigma$?
1. Consider removing the line about logistic regression from your abstract. You don't really provide any new guarantee there.
1. You should mention in the body of the paper that there are experiments in the appendix.
1. The appendix is ordered in a hard-to-read way. I constantly found myself flipping back-and-forth between pages to remember what Lemma X.X refers to. Further, finding the proof of a specific lemma takes a while. Lastly, the order of proofs makes it hard to jump to the proofs that interest me. I would recommend a few changes:
    - At the beginning of *every* proof in the appendix, fully restate the lemma/theorem/corollary.
    - When you state a lemma without proof, link to its proof.
    - List all defined symbols at the beginning or end of the appendix. There's a lot of symbols, and some like $\varepsilon_\lambda$ are defined within some lemma statements, which is easy to skip over on accident.
    - Consider just reordering the entire appendix to be fully bottom-up? Just start with the lemmas that requires no other lemmas, and build up from there?
1. All experiments in this paper would benefit from error bars. Consider highlighting the area between $25^{th}$ and $75^{th}$ quartiles on each plot? Figure 2(d) at $\\|\mu\\|_2^2 = 4,6$ might be a bit non-linear, and some error bars would dissuade my concern, for example. It would also resolve the weird bump at $\\|\mu\\|=2,\alpha=0.8$ on Figure 2(b).

Some typos and smaller confusions:
1. [Line 64] "our result shows that to achieve $o(1)$ population risk" If $n$ is constant, what is this little-o depending on? $d$?
1. [Line 293] The details are in Appendix A.1, not Appendix 4.1.
1. [Line 319] Consider trying to unify the $Q=Z\Lambda^{1/2}V^\intercal$ notation with the content of [Line 120]. Maybe introduce the $Z$ notation there as an equivalent characterization? It's a little confusing as written, since this notation hasn't been introduced earlier, so it takes extra effort to verify. It would be more clear if I thought about this characterization when $Q$ was first defined and fresher in my memory.
1. [Line 605] Consider justifying $\\|\mu\\|_2^2 \geq \mu^\intercal Q^\intercal (QQ^\intercal)^{-1} Q\mu$, which I believe follows from $Q^\intercal (QQ^\intercal)^{-1} Q$ being a hat matrix?
1. [Line 612] I don't think $v_i$ was defined anywhere.
1. [Line 642] Unmatched parenthesis in the definition of $J_2$.
1. [Line 733] "the dependency of" should be removed
1. [Line 747] Should read "holds" not "hold"

**Time Spent Reviewing:**

7

---

> ### Author Response · Authors · 2021-08-10
> **Response to Reviewer 1ykL**
>
> Thank you for your constructive review. We address your comments as follows.
>
> 1. How fundamental is the covariance matrix's trace lower bound assumption(s)? Why is trace meaningful here?
>
> Our analysis is based on certain concentration inequalities for the data Gram matrix, and the eigenvalues of the Gram matrix concentrates around $\mathrm{tr}( \mathbf{\Sigma} )$ (Lemma A.4). Intuitively, $\mathrm{tr}( \mathbf{\Sigma} )$ serves as a natural surrogate of the dimension in the anisotropic setting, which enables us to study the infinite-dimensional maximum margin classifier. This is the main reason that our theorems require such conditions on the trace of $\mathbf{\Sigma}$. Assuming lower bounds of $\mathrm{tr}( \mathbf{\Sigma} )$ ensures that we are indeed studying the over-parameterized setting (e.g., $d \gg n$).
>
> 2. What kinds of covariance matrices fail the conditions on $\mathrm{tr}( \mathbf{\Sigma} )$ in Theorem 3.1? Do they have a flat spectrum? Does the sample complexity n limit how flat the spectrum can be?
> (For example, Theorem 3.1 requires $\mathrm{tr}( \mathbf{\Sigma} ) / || \mathbf{\Sigma} ||_F \geq n$ amongst other constraints. Is this a reasonable way to interpret the hard constraint? What does this really say about $\mathbf{\Sigma}$?)
>
> Given that the dimension $d$ is large enough, and that the largest eigenvalue of $\mathbf{\Sigma}$ is of constant order, the flatter the spectrum is, the easier for the condition in Theorem 3.1 to be satisfied. An example when the condition fails is when the eigenvalues of $\mathbf{\Sigma}$ decay very fast. For example, when $\lambda_k = k^{-\alpha}$ with $\alpha > 1$, the condition cannot be satisfied for large $n$.
>
> It is true that part of the condition can be interpreted as $\mathrm{tr}( \mathbf{\Sigma} ) / || \mathbf{\Sigma} ||_F \geq n $. Note that $ \mathrm{tr}( \mathbf{\Sigma} ) / || \mathbf{\Sigma} ||_F $ achieves its maximum when all eigenvalues of $\mathbf{\Sigma}$ are equal. Moreover, when all eigenvalues are approximately equal,  $ \mathrm{tr}( \mathbf{\Sigma} ) / || \mathbf{\Sigma} ||_F $ is of order $\sqrt{d}$, in which case the condition can be interpreted as $d > n^2$.
>
> 3. 4 and 5, Suggestions for the revision:
> - Remove the line about logistic regression from the abstract.
> - Mention in the body of the paper that there are experiments in the appendix.
> - Presentation of the proof in the appendix.
>
> Thank you for your suggestions. We will revise the paper accordingly, and improve the presentation of the proof in the appendix.
>
> 6. Add error bars in the experiments.
>
> Thank you for your suggestion. We will add the error bar in the plots.
>
> 7.  Typos and smaller confusions
>
> Thank you for pointing these out. We will fix them in the final version.

---

### Official Review · Reviewer_1YJ4 · 2021-07-16

**Rating:** 6
**Confidence:** 4

**Summary:**

The paper studies risk bounds for the hard-margin SVM binary classification problem in the over-parametrized regime, where the number of samples is much smaller than the dimension of the instance space. It is assumed that the samples follow a distribution corresponding to a mixture of two sub-gaussian components, one component for each label. These components are assumed to have the same covariance matrix Sigma and opposing means mu and -mu. The main results of the paper are upper and lower bounds for the risk of the SVM classifier as trained on the data. By previous results, this also has implications to the minimum norm solution learned by gradient decent. The derived bounds are tighter than previous bounds and extend to the anisotropic setting. The lower bound derived matches the upper bound in some regimes of the parameters, suggesting optimality. They also discuss in some detail the role played by the eigenvalues of Sigma in the risk bounds, showing a nice separation to two regimes.

**Ethical Concerns:**

None.

**Limitations And Societal Impact:**

Yes.

**Main Review:**

The paper is well written and easy to follow and the results seem technically correct. The results, however, are somewhat incremental as compared to previous work on the subject, hence my relatively low score.

Some questions/comments:

- In the introduction, the results are presented without the dependence on $\Sigma$, stating rates that are not scale invariant as one would expect.

- You assume that the distribution is centered. How does your bounds change when one also needs to estimate the center of the mixture, or equivalently, learn an additional bias parameter in the SVM classifier.

- Please give the formal definition of the sub-gaussian/exponential norms, or at least state an appropriate reference where they are mentioned (line 104).

- Is the condition $||\mu||_2 \geq C||\Sigma||_2$ in Lemma 4.4 correct? It is not scale invariant.


**Time Spent Reviewing:**

4

---

> ### Author Response · Authors · 2021-08-10
> **Response to Reviewer 1YJ4**
>
> Thank you for your helpful comments. Our detailed responses to your questions are as follows.
>
> 1. The results are somewhat incremental as compared to previous work on the subject
>
> Compared with previous work, our paper gives a tighter risk upper bound, and gives the first lower bound of its kind, which demonstrates that our risk upper bound is tight. Moreover, our result covers the anisotropic setting, and is proved based on certain novel proof techniques, including the application of the polarization identity to establish equivalence between the maximum margin classifier and the minimum norm interpolator.
>
> 2. In the introduction, the presented results are not scale invariant.
>
> In the introduction, for simplicity, the bounds mentioned in the first bullet of contributions (lines 50-56) are for the setting where $\mathbf{\Sigma} = \mathbf{I}$. Therefore, you can roughly treat $d$ as $|| \mathbf{\Sigma} ||_F^2$. Based on this, you can see that these bounds here are indeed scale invariant, as scaling the input $\mathbf{x}$ does not change the results.
>
> In terms of the conditions presented in the third bullet (lines 60-68), they are derived for the specific setting where $\lambda_k = k^{-\alpha}$. In other words, we directly assume the scale of $\mathbf{\Sigma}$. Therefore these conditions are not meant to be scale invariant.
>
> We will clarify this in the revised version.
>
> 3. How do the bounds change when the data model is not centered, and the SVM classifier has an additional bias parameter.
>
> When the data model is not centered, we can either center the data using the sample mean, or introduce a bias term in the maximum margin classifier. In both cases, the analysis will be a little more involved, but we believe it won’t change the results significantly, and the rates in the risk bounds should remain the same.
>
> 4. Formal definition of the sub-gaussian/exponential norms. Adding related references
>
> Thanks for pointing it out. We will give the formal definition of sub-gaussian/exponential norms in the revision, and add the related references.
>
> 5. Is the condition $|| \mathbf{\mu} ||_2 \geq C || \mathbf{\Sigma} ||_2 $ in Lemma 4.4 correct? It is not scale invariant.
>
> Thanks for pointing it out. This is a typo. This condition should be $|| \mathbf{\mu} ||_2 \geq C || \mathbf{\mu} ||\_{ \mathbf{\Sigma} } $. We will fix it in the revision. Note that this is only a typo in the lemma statement, and there is no need to change the proof, as the current proof of Lemma 4.4 is only based on the assumption that $|| \mathbf{\mu} ||_2 \geq C || \mathbf{\mu} ||\_{\mathbf{\Sigma}} $ (line 527 in the supplementary material).

---

> > ### Comment · Reviewer_1YJ4 · 2021-08-10
> > **Corrections**
> >
> > Regarding point 5, in your "corrected" condition above, the LHS should be squared, $\lVert \mu \rVert_2^2 \geq C \lVert \mu \rVert_\Sigma$?
> >
> > Regarding point 3, are there any known results on estimating the mean vector in the regime considered in this paper (e.g., $d\geq n^2$ in the isotropic case)?

---

> > > ### Author Response · Authors · 2021-08-11
> > > **Thank you for your further comment and question.**
> > >
> > > Thank you for your further comment and question.
> > >
> > > 1. Regarding point 5, in your "corrected" condition above, the LHS should be squared, $|| \mathbf{\mu} ||_2^2 \geq C || \mathbf{\mu} ||\_{\mathbf{\Sigma}}$?
> > >
> > > Sorry for the typo. It should be $|| \mathbf{\mu} ||_2^2 \geq C || \mathbf{\mu} ||\_{\mathbf{\Sigma}} $, as is used in line 527 in the supplementary material. To verify that this condition $|| \mathbf{\mu} ||_2^2 \geq C || \mathbf{\mu} ||\_{\mathbf{\Sigma}} $ is indeed scale invariant: Under our model assumption, $\mathbf{x} = \mathbf{\mu} + \mathbf{q}$, and $\mathbf{\Sigma} = \mathbb{E} [  \mathbf{q}  \mathbf{q}^\top ]$. Therefore when $\mathbf{x} $ is multiplied by $c$, $\mathbf{\mu}$ is also scaled by $c$, and $\mathbf{\Sigma}$ is scaled by $c^2$. Therefore the LHS $|| \mathbf{\mu} ||_2^2$ is scaled by $c^2$. Meanwhile, the RHS $C || \mathbf{\mu} ||\_{\mathbf{\Sigma}} = C \sqrt{ \mathbf{\mu}^\top \mathbf{\Sigma} \mathbf{\mu} }$ is also scaled by $c^2$.
> > >
> > >
> > > 2. Regarding point 3, are there any known results on estimating the mean vector in the regime considered in this paper (e.g., $d \geq n^2$  in the isotropic case)?
> > >
> > > We would like to clarify that we do not need to bound the estimation error of the sample mean estimator directly in that regime. Instead, we only need to keep a “sample average” term when deriving the centered SVM predictor, and then bound the risk jointly.
> > >
> > > Here is some intuition. Suppose that the labels $y = +1$ and $y = -1$ correspond to the Gaussian distributions with mean vectors $\mathbf{\mu}_1$ and $\mathbf{\mu}_2$ respectively, and let $\tilde{\mathbf{\mu}} =( \mathbf{\mu}_1 - \mathbf{\mu}_2 ) / 2$. For simplicity, let us assume here that there are exactly $n/2$ data points from each class. Then, we can consider a centered SVM on the modified dataset with the $i$-th input being
> > >
> > > $\tilde{\mathbf{x}}_i := \mathbf{x}_i - \frac{1}{n} \sum\_{i'=1}^n \mathbf{x}\_{i'} = y\_i \cdot \tilde{\mathbf{\mu}} + \mathbf{q}\_i - \frac{1}{n} \sum\_{i'=1}^n \mathbf{q}\_{i'}  =  y\_i \cdot \tilde{\mathbf{\mu}} + \tilde{\mathbf{q}}\_i.$
> > >
> > > From the above calculation, we can see that the distribution of the training data points $\\{ \tilde{\mathbf{x}}_i \\}\_{i=1}^n$ can still be expressed in a similar form as in the paper. The major difference is that in this new setting, $\tilde{\mathbf{q}}\_{i}$ and $\tilde{\mathbf{q}}\_{i'}$ are correlated and no longer independent even when $ i \neq i’ $. Nevertheless, many of the key lemmas in our analysis such as Lemma A.4 and Lemma A.5 can still hold because they actually do not require such independence. Some other lemmas may need to be reproved with extra efforts.
> > >
> > > This setting is definitely interesting but beyond the focus of our paper. We are happy to briefly comment on it in the future work section.

---

### Official Review · Reviewer_nXks · 2021-07-18

**Rating:** 7
**Confidence:** 4

**Summary:**

This paper proves new risk bounds for overfitting in the over-parameterized linear classification setting when the data is generated by a sub-Gaussian mixture. In certain regimes, the results show that overfitting can be benign, that is, overfitting still leads to a good test error. An important property of the proposed upper bound on the risk is that it is not directly dependent on the dimension, but rather on certain norms of the covariance matrix. This is useful because, as shown in the paper, if the eigenvalues of the covariance matrix decay fast enough, then requirements for benign overfitting become dimension independent.

**Limitations And Societal Impact:**

Please see the main review.

**Main Review:**

This paper proves new risk bounds for overfitting in the over-parameterized linear classification setting when the data is generated by a sub-Gaussian mixture. In certain regimes, the results show that overfitting can be benign, that is, overfitting still leads to a good test error. An important property of the proposed upper bound on the risk is that it is not directly dependent on the dimension, but rather on certain norms of the covariance matrix. This is useful because, as shown in the paper, if the eigenvalues of the covariance matrix decay fast enough, then requirements for benign overfitting become dimension independent.

This paper is good step towards understanding benign overfitting, even though the paper only considers linear classification in sub-Gaussian mixture setting. Corollary 3.5 proves that even if the eigenvalues of the covariance matrix do not decay very fast, the requirement for small test error scale sub-linearly with dimension. In particular, as long as the the number of samples is a large enough constant, and the means of the classes are separated by $d^{1/4}$, the risk of maximum margin classifier is $o(1)$.

Overall, the paper is well written, and the usefulness of the theorem is well illustrated with examples. The intuition is explained well and the theorem proof in the main paper is also helpful.

My first concern about the results is that the requirement $\text{tr}(\mathbf{\Sigma})\geq C \max (n^{3/2}||\mathbf{\Sigma}||_2,n||\mathbf{\Sigma}||_F)$, which in the isotropic Gaussian case implies that the theorem requires $d\geq n^2$. Thus, the result is only applicable to extremely over-parameterized setting. Is this requirement a necessary condition for such results? In the mild over-parameterized setting (say $d\geq 100 n$), would overfitting no-longer be benign? This can be the case, for example, if we keep the data distribution fixed and keep increasing the number of samples $n$.

Another concern that I have is whether these results can be transferred to the setting of deep learning. In the linear classification setting considered in this paper, the results are essentially for the maximum margin classifier, with the motivation being that asymptotically, SGD converges to the maximum margin classifier. This is not the case for deep learning, where if we just benignly overfit the data, then there still are adversarial examples. Hence, overfitting in deep learning does not lead to maximum margin classifiers usually. Given that benign overfitting still leads to a good test accuracy for deep networks, an important question is whether the explanation offered by this paper is still valid for deep learning.

**Time Spent Reviewing:**

5

---

> ### Author Response · Authors · 2021-08-10
> **Response to Reviewer nXks**
>
> Thank you for your insightful comments. Below are our answers to your questions.
>
> 1. Theorems are for the extremely over-parameterized setting ($d>n^2$ for isotropic data). Is it a necessary condition?
>
> It is true that for isotropic Gaussians our theorem requires $d > n^2$. This is partly due to our proof technique that relies on the equivalence between the maximum margin classifier and the minimum norm interpolator. To our knowledge, our condition is among the mildest in the existing results on benign overfitting of maximum margin classifiers on sub-Gaussian mixture data. However, $d>n^2$ may not be a necessary condition for benign overfitting. In fact, we tend to believe that benign-overfitting can still occur under a mild over-parameterized setting (say,
> d≥100n). In order to show this, a different proof technique that does not rely on the equivalence result may be needed.
>
> 2. Extension to the setting of deep learning
>
> Our current result is only for linear models and relies on the maximum margin predictor in the parameter space. It serves as the first step to study more complicated models such as neural networks, and we hope that the insights and some of the mathematical analysis can be carried over to deep learning. Given the empirically observed benign overfitting phenomenon in deep learning, there is a hope that similar results can be proved for deep learning.  In fact, for neural networks with homogeneous activation functions, it has been proved in [16] that gradient descent minimizing the logistic loss will also converge to the maximum margin solution, or more precisely a KKT point of a maximum margin problem. Therefore, intuitively speaking, the extension of our results to deep learning can potentially rely on the maximum margin predictor (minimum-norm interpolator) defined in neural network function space instead of the linear function space.

---

> > ### Comment · Reviewer_nXks · 2021-08-24
> > **Response to author response**
> >
> > Thank you for addressing my concerns. I am keeping my score.

---

### Official Review · Reviewer_GwQP · 2021-07-18

**Rating:** 6
**Confidence:** 4

**Summary:**

This paper studied the classification error of max margin linear classifier in high dimensional Sub-Gaussian mixtures. Tight upper and lower bounds are derived, which generalizes the prior works on Gaussian Mixtures.


**Ethical Concerns:**

-

**Limitations And Societal Impact:**

-

**Main Review:**

This paper studied the classification error of max margin linear classifier in high dimensional Sub-Gaussian mixtures. This is an important problem, due to its connection to the implicit bias of first order algorithms, interpolation and the double descent phenomena.

Tight upper and lower bounds are derived, which generalizes the prior works on Gaussian Mixtures. Overall, this paper is well-written, the results are solid and I enjoyed reading it. A few comments are in order:

(1) It might be better to expand the prior works section to include some relevant papers on high dimensional Gaussian Mixture. For example, the exp(- |mu|^4/(|mu|^2+d/n)) -type risk bounds also appeared in high dimensional Gaussian clustering literature, e.g.

https://arxiv.org/pdf/1812.08078.pdf

https://arxiv.org/pdf/2006.14062.pdf

and many other references therein.



(2) A major limitation I see from this work, is that its analysis heavily relies on the equivalence between max margin classifier and min-norm interpolator, see [19]. As the authors already mentioned in this paper (e.g. corollary 3.3), for example, in the text book example of isotropic Gaussian mixture with a constant separation, this can only happen when d > n^2, or the number of samples n is at most sqrt(d). In contrast, the n/d = constant setting is much more standard, and in certain sense more interesting. In my opinion, the results in this paper are far from a complete picture about max margin classifier - In other words, they are really about min-norm interpolator, which happens to be equivalent to max-margin classifier in the small sample regime (and therefore the title is a little bit misleading).

Overall, my recommendation is weak accept - any comments and clarifications from the authors are welcome.

**Time Spent Reviewing:**

4

---

> ### Author Response · Authors · 2021-08-10
> **Response to Reviewer GwQP**
>
> Thank you for your supportive comments. We address your questions as follows.
>
> 1. Related work on high dimensional Gaussian Mixtures.
>
> We appreciate your suggestion. We will discuss these relevant papers in the revision.
>
> 2. Reliance on the equivalence result. The $d > n^2$ condition under the isotropic setting... far from a complete picture about max margin classifier
>
> It is true that our analysis relies on the equivalence between the maximum margin classifier and the minimum norm interpolator, and we need $d>n^2$ in the isotropic Gaussian mixture setting. Nevertheless, our condition is still among the mildest in existing results on benign overfitting of maximum margin classifiers on sub-Gaussian mixtures.
>
> We agree that $n/d = \mathrm{constant}$ is an interesting setting. However, we believe that the setting $d = \omega(n)$ is equally interesting, as it represents the heavily over-parameterized case. We are aware of some recent papers studying the $n/d = \mathrm{constant}$ setting under the assumption that the input data are generated from a single Gaussian distribution (e.g., arXiv:1911.01544). That analysis relies on subtle properties of random matrices. It is not clear whether a similar analysis can be applied to a much broader class of sub-Gaussian mixtures. Indeed, a complete picture that unifies the $n/d = \mathrm{constant}$ setting and the $d = \omega(n)$ setting is still missing. Our paper does not aim to give such a unified theory. Instead, we aim at giving a tight characterization of the risk for the maximum margin classifier in the $d = \omega(n)$ setting. We will comment on the $n/d = constant$ setting in the revision and leave it as a future work direction.

---

### Decision · Program_Chairs · 2021-09-27

**Decision:**

Accept (Poster)

**Comment:**

This paper is favored by two reviewers and is acceptable by all. Reviewers agree that it is clearly written and organized, even enjoyable to read, and that the results are solid on their own. Throughout the various exchanges between reviewers and authors, some valid suggestions emerged that may strengthen the paper, for instance:

* Discussing the possibility that d > n^2 is required and situating that relative to assumptions in related work (from discussion with GwQP and nXks), namely highlighting the sense in which this is comparatively mild.

* Considering commenting on how to approach an extension to data that is not centered (from discussion with 1YJ4). This may be more of an optional suggestion, but again may be useful to understanding the setup and analysis.

Overall the paper makes a number of technical contributions to the actively developing topic of studying benign overfitting. Together with mostly favorable reviews, I recommend it for acceptance.